# Rhino: Deep Causal Temporal Relationship Learning with history-dependent noise

**Wenbo Gong, Joel Jennings, Cheng Zhang & Nick Pawlowski**
Microsoft Research
Cambridge, UK
`{wenbogong, joeljennings, cheng.zhang, nick.pawlowski}`
`@microsoft.com`

## Abstract

Discovering causal relationships between different variables from time series data has been a long-standing challenge for many domains such as climate science, finance, and healthcare. Given the complexity of real-world relationships and the nature of observations in discrete time, causal discovery methods need to consider non-linear relations between variables, instantaneous effects and history-dependent noise (the change of noise distribution due to past actions). However, previous works do not offer a solution addressing all these problems together. In this paper, we propose a novel causal relationship learning framework for time-series data, called Rhino, which combines vector auto-regression, deep learning and variational inference to model non-linear relationships with instantaneous effects while allowing the noise distribution to be modulated by historical observations. Theoretically, we prove the structural identifiability of Rhino. Our empirical results from extensive synthetic experiments and two real-world benchmarks demonstrate better discovery performance compared to relevant baselines, with ablation studies revealing its robustness under model misspecification.

## 1 Introduction

Time series data is a collection of data points recorded at different timestamps describing a pattern of chronological change. Identifying the causal relations between different variables and their interactions through time (Spirtes et al., 2000; Berzuini et al., 2012; Guo et al., 2020; Peters et al., 2017) is essential for many applications e.g. climate science, health care, etc. Randomized control trials are the gold standard for discovering such relationships, but may be unavailable due to cost and ethical constraints. Therefore, causal discovery with just observational data is important and fundamental to many real-world applications (Löwe et al., 2022; Bussmann et al., 2021; Moraffah et al., 2021; Wu et al., 2020; Runge, 2018; Tank et al., 2018; Hyvärinen et al., 2010; Pamfil et al., 2020).

The task of temporal causal discovery can be challenging for several reasons: (1) relations between variables can be non-linear in the real world; (2) with a slow sampling interval, everything happens in between will be aggregated into the same timestamp, i.e. instantaneous effect; (3) the noise may be non-stationary (its distribution depends on the past observations), i.e. history-dependent noise. For example, in stock markets, the announcements of some decisions from a leading company after the market closes may have complex effects (i.e. non-linearity) on its stock price immediately after the market opening (i.e. slow sampling interval and instantaneous effect) and its price volatility may also be changed (i.e. history-dependent noise). Similarly, in education, students that recently earned good marks on algebra tests should also score well on an upcoming algebra exam with little variation (i.e. history-dependent noise).

To the best of our knowledge, existing frameworks' performances suffer in many real-world scenarios as they cannot address these aspects in a satisfactory way. Especially, history-dependent noise has been rarely considered in past. A large category of the preceding works, called *Granger causality* (Granger, 1969), is based on the fact that cause-effect relationships can never go against time. Despite many recent advances (Wu et al., 2020; Shojaie & Michailidis, 2010; Siggiridou &

Kugiumtzis, 2015; Amornbunchornvej et al., 2019; Löwe et al., 2022; Tank et al., 2018; Bussmann et al., 2021; Dang et al., 2018; Xu et al., 2019), they all rely on the absence of instantaneous effects with a fixed noise distribution. Constraint-based methods have also been extended for time series causal discovery (Runge, 2018; 2020), which is commonly applied by folding the time-series. This introduced new assumptions and translated the aforementioned requirements to challenges in conditional independence testing (Shah & Peters, 2020). Additionally, they require a stronger faithfulness assumption and can only identify the causal graph up to a Markov equivalence class without detailed functional relationships.

An alternative line of research leverages the development of causal discovery with functional causal models (Hyvärinen et al., 2010; Pamfil et al., 2020; Peters et al., 2013). They can model both instantaneous and lagged effects as long as they have theoretically guaranteed *structural identifiability*. Unfortunately, they do not consider history-dependent noise. One central challenge of modelling this dependency is that noise depending on the lagged parents may break the model structural identifiability. For static data, Khemakhem et al. (2021) proves the structural identifiability only when this dependency is restricted to a simple functional form. Thus, the key research question is whether the identifiability can be preserved with complex historical dependencies in the temporal setting.

Motivated by these requirements, we propose a novel temporal discovery framework called Rhino (*deep causal temporal relationship learning with history dependent noise*), which can model non-linear lagged and instantaneous effects with flexible history-dependent noise. Our contributions are:

- A novel causal discovery framework called Rhino, Revision(Q2)-Reviewer hiTaconsisting of a novel functional form of its SEMs and variational training framework, where the proposed form of its SEM combines vector auto-regression and deep learning to model non-linear lagged and instantaneous effects with history-dependent noise.

- We prove that Rhino SEMs with the proposed form are structurally identifiable. To achieve this, we provide general conditions for structural identifiability with history-dependent noise, of which the form of Rhino SEMs is a special case. Furthermore, we clarify relations to several previous works.

- We conduct extensive synthetic experiments with ablation studies to demonstrate the advantages of Rhino and its robustness across different settings. Additionally, we compare its performance to a wide range of baselines in two real-world discovery benchmarks.

## 2 BACKGROUND

In this section, we briefly introduce necessary prerequisite knowledge. In particular, we focus on structural equation models, Granger causality (Granger, 1969) and vector auto-regression. For review of more recent related work, please refer to Section 5.

**Structural Equation Models (SEMs)** Consider $\boldsymbol{X} \in \mathbb{R}^D$ with $D$ variables, SEM describes the causal relationships between them given a causal graph $\boldsymbol{G}$:

$$X^i = f_i(\mathbf{Pa}_G^i, \epsilon^i) \tag{1}$$

where $\mathbf{Pa}_G^i$ are the parents of node $i$ and $\epsilon^i$ are mutually independent noise variables. Under the context of multivariate time series, $\boldsymbol{X}_t = \left(X_t^i\right)_{i \in \boldsymbol{V}}$ where $\boldsymbol{V}$ is a set of nodes with size $D$, the corresponding SEM given a temporal causal graph $\boldsymbol{G}$ is

$$X_t^i = f_{i,t}(\mathbf{Pa}_G^i(< t), \mathbf{Pa}_G^i(t), \epsilon_t^i), \tag{2}$$

where $\mathbf{Pa}_G^i(< t)$ contains the parent values specified by $G$ in previous time (*lagged parents*); $\mathbf{Pa}_G^i(t)$ are the parents at the current time $t$ (*instantaneous parents*). The above SEM induces a joint distribution over the stationary time series $\{\boldsymbol{X}_t\}_{t=0}^T$ (see Assumption 1 in Appendix B for the definition). However, functional causal models with the above general form cannot be directly used for causal discovery due to the structural unidentifiability (Lemma 1, Zhang et al. (2015) One way to solve this is sacrificing the flexibility by restricting the functional class. For example, additive noise models (ANM), (Hoyer et al., 2008)

$$X^i = f_i(\mathbf{Pa}_G(X^i)) + \epsilon_i, \tag{3}$$

which have recently been used for causal reasoning with non-temporal data (Geffner et al., 2022).

**Granger Causality** Granger causality (Granger, 1969) has been extensively used for temporal causal discovery. It is based on the idea that the series $\boldsymbol{X}^j$ does not Granger cause $\boldsymbol{X}^i$ if the history, $\boldsymbol{X}^j_{<t}$, does not help the prediction of $X^i_t$ for some $t$ given the past of all other time series $\boldsymbol{X}^k$ for $k \neq j, i$.

**Definition 2.1** (Granger Causality (Tank et al., 2018; Löwe et al., 2022))**.** Given a multivariate stationary time series $\{\boldsymbol{X}_t\}^T_{t=0}$ and a SEM $f_{i,t}$ defined as

$$X^i_t = f_{i,t}(\mathbf{Pa}^i_G(< t)) + \epsilon^i_t, \tag{4}$$

$\boldsymbol{X}^j$ Granger causes $\boldsymbol{X}^i$ if $\exists l \in [1, t]$ such that $X^j_{t-l} \in \mathbf{Pa}^i_G(< t)$ and $f_{i,t}$ depends on $X^j_{t-l}$.

Granger causality is equivalent to causal relations for *directed acyclic graph* (*DAG*) if there are no latent confounders and instantaneous effects (Peters et al., 2013; 2017). Apart from the lack of instantaneous effects, it also ignore the history-dependent noise with independent $\epsilon^i_t$.

**Vector Auto-regressive Model** Another line of research focuses on directly fitting the identifiable SEM to the observational data with instantaneous effects. One commonly-used approach is called vector auto-regression (Hyvärinen et al., 2010; Pamfil et al., 2020):

$$X^i_t = \beta^i + \sum_{\tau=0}^{K} \sum_{j=1}^{D} B_{\tau,ji} X^j_{t-\tau} + \epsilon^i_t \tag{5}$$

where $\beta^i$ is the offset, $\tau$ is the model lag, $\boldsymbol{B}_\tau \in \mathbb{R}^{D \times D}$ is the weighted adjacency matrix specifying the connections at time $t - \tau$ (i.e. if $B_{\tau,ji} = 0$ means no connection from $X^j_{t-\tau}$ to $X^i_t$) and $\epsilon^i_t$ is the independent noise. Under these assumptions, the above linear SEM is structurally identifiable, which is a necessary condition for recovering the ground truth graph (Hyvärinen et al., 2010; Peters et al., 2013; Pamfil et al., 2020). However, the above linear SEM with independent noise variables is too restrictive to fulfil the requirements described in Section 1. Therefore, the research question is how to design a structurally identifiable non-linear SEM with flexible history-dependent noise.

## 3 RHINO: RELATIONSHIP LEARNING WITH HISTORY DEPENDENT NOISE

This section introduces Rhino: Section 3.1 describes the novel functional form for Rhino SEMs, allowing for history-dependent noise. Section 3.2 details the associated variational inference framework for causal discovery.

### 3.1 MODEL FORMULATION

For a multivariate stationary time series $\{\boldsymbol{X}_t\}^T_{t=0}$, we assume that their causal relations follow a temporal adjacency matrix $\boldsymbol{G}_{0:K}$ with maximum lag $K$ where $\boldsymbol{G}_{\tau \in [1,K]}$ specifies the lagged effects between $\boldsymbol{X}_{t-\tau}$ and $\boldsymbol{X}_t$, $\boldsymbol{G}_0$ specifies the instantaneous parents. We define $G_{\tau,ij} = 1$ if $X^i_{t-\tau} \to X^j_t$ and 0 otherwise. [1] We propose a novel functional form for Rhino's SEM that incorporates non-linear relations, instantaneous effects, and flexible history-dependent noise:

$$X^i_t = f_i(\mathbf{Pa}^i_G(< t), \mathbf{Pa}^i_G(t)) + g_i(\mathbf{Pa}^i_G(< t), \epsilon^i_t) \tag{6}$$

where $f_i$ is a general differentiable non-linear function, and $g_i$ is a differentiable transform s.t. the transformed noise has a proper density. Despite of an additive structure, our formulation offers much more flexibility in both functional relations and noise distributions compared to previous works (Pamfil et al., 2020; Peters et al., 2013). By placing few restrictions on $f_i, g_i$, it can capture functional non-linearity through $f_i$ and transform $\epsilon^i_t$ through a flexible function $g_i$, depending on $\mathbf{Pa}^i_G(< t)$, to capture the history dependency of the additive noise.

Next, we propose flexible functional designs for $f_i, g_i$, which must respect the relations encapsulated in $\boldsymbol{G}$. Namely, if $X^j_{t-\tau} \notin \mathbf{Pa}^i_G(< t) \cup \mathbf{Pa}^i_G(t)$, then $\partial f_i / \partial X^j_{t-\tau} = 0$ and similarly for $g_i$. We design

$$f_i(\mathbf{Pa}^i_G(< t), \mathbf{Pa}^i_G(t)) = \zeta_i \left( \sum_{\tau=0}^{K} \sum_{j=1}^{D} G_{\tau,ji} \ell_{\tau j} \left( X^j_{t-\tau} \right) \right) \tag{7}$$

---

[1] In the following, we interchange the usage of the notation $\boldsymbol{G}$ and $\boldsymbol{G}_{0:K}$ for brevity.

where $\zeta_i$ and $\ell_{\tau i}$ ($i \in [1, D]$ and $\tau \in [0, K]$) are neural networks. For efficient computation, we use weight sharing across nodes and lags: $\zeta_i(\cdot) = \zeta(\cdot, \boldsymbol{u}_{0,i})$ and $\ell_{\tau j}(\cdot) = \ell(\cdot, \boldsymbol{u}_{\tau,j})$, where $\boldsymbol{u}_{\tau,i}$ is the trainable embedding for node $i$ at time $t - \tau$.

The design of $g_i$ needs to properly balance the flexibility and tractability of the transformed noise density. We choose a conditional normalizing flow, called conditional spline flow (Trippe & Turner, 2018; Durkan et al., 2019; Pawlowski et al., 2020), with a fixed Gaussian noise $\epsilon_t^i$ for all $t$, $i$. The spline parameters are predicted using a hyper-network with a similar form to Eq. (7) to incorporate history dependency. The only difference is now $\tau$ is summed over $[1, K]$ to remove the instantaneous parents. Due to the invertibility of $g_i$, the noise likelihood conditioned on lagged parents is

$$p_{g_i}(g_i(\epsilon_t^i)|\mathbf{Pa}_G^i(<t)) = p_\epsilon(\epsilon_t^i) \left| \frac{\partial g_i^{-1}}{\partial \epsilon_t^i} \right|. \tag{8}$$

## 3.2 Variational Inference for causal discovery

Rhino adopts a Bayesian view of causal discovery (Heckerman et al., 2006), which aims to learn a graph posterior distribution instead of inferring a single graph. For $N$ observed multivariate time series $\boldsymbol{X}_{0:T}^{(1)}, \dots, \boldsymbol{X}_{0:T}^{(N)}$, the joint likelihood with model parameter $\theta$ is

$$p(\boldsymbol{X}_{0:T}^{(1)}, \dots, \boldsymbol{X}_{0:T}^{(N)}, \boldsymbol{G}) = p(\boldsymbol{G}) \prod_{n=1}^{N} p_\theta(\boldsymbol{X}_{0:T}^{(n)}|\boldsymbol{G}). \tag{9}$$

**Graph Prior** When designing the graph prior, we combine three components: (1) DAG constraint; (2) graph sparseness prior; (3) domain-specific prior knowledge (optional). Inspired by the NOTEARS (Zheng et al., 2018; Geffner et al., 2022; Morales-Alvarez et al., 2021), we propose the following unnormalised prior

$$p(\boldsymbol{G}) \propto \exp\left(-\lambda_s \|\boldsymbol{G}_{0:K}\|_F^2 - \rho h^2(\boldsymbol{G}_0) - \alpha h(\boldsymbol{G}_0) - \lambda_p \|\boldsymbol{G}_{0:K} - \boldsymbol{G}_{0:K}^p\|_F^2\right) \tag{10}$$

where $h(\boldsymbol{G}) = \text{tr}(e^{\boldsymbol{G} \odot \boldsymbol{G}}) - D$ is the DAG penalty proposed in (Zheng et al., 2018) and is 0 if and only if $\boldsymbol{G}$ is a DAG; $\odot$ is the Hadamard product; $\boldsymbol{G}^p$ is an optional domain-specific prior graph, which can be used when partial domain knowledge is available; $\lambda_s, \lambda_p$ specify the strength of the graph sparseness and domain-specific prior terms respectively; $\alpha, \rho$ characterize the strength of the DAG penalty. Since the lagged connections specified in $\boldsymbol{G}_{1:K}$ can only follow the direction of time, only the instantaneous part, $\boldsymbol{G}_0$, can contain cycles. Thus, the DAG penalty is only applied to $\boldsymbol{G}_0$.

**Variational Objective** Unfortunately, the exact graph posterior $p(\boldsymbol{G}|\boldsymbol{X}_{0:T}^{(1)}, \dots, \boldsymbol{X}_{0:T}^{(N)})$ is intractable due to the large combinatorial space of DAGs. To overcome this challenge, we adopt variational inference (Blei et al., 2017; Zhang et al., 2018), which uses a variational distribution $q_\phi(\boldsymbol{G})$ to approximate the true posterior. We choose $q_\phi(\boldsymbol{G})$ to be a product of independent Bernoulli distributions (refer to Appendix E for details). The corresponding *evidence lower bound* (*ELBO*) is

$$\log p_\theta\left(\boldsymbol{X}_{0:T}^{(1)}, \dots, \boldsymbol{X}_{0:T}^{(N)}\right) \geq \underbrace{\mathbb{E}_{q_\phi(\boldsymbol{G})}\left[\sum_{n=1}^{N} \log p_\theta(\boldsymbol{X}_{0:T}^{(n)}|\boldsymbol{G}) + \log p(\boldsymbol{G})\right] + H(q_\phi(\boldsymbol{G}))}_{\text{ELBO}(\theta,\phi)} \tag{11}$$

where $H(q_\phi(\boldsymbol{G}))$ is the entropy of $q_\phi(\boldsymbol{G})$. From the causal Markov assumption and auto-regressive nature, we can further simplify

$$\log p_\theta(\boldsymbol{X}_{0:T}^{(n)}|\boldsymbol{G}) = \sum_{t=0}^{T} \sum_{i=1}^{D} \log p_\theta(X_t^{i,(n)}|\mathbf{Pa}_G^i(<t), \mathbf{Pa}_G^i(t)) \tag{12}$$

and from Rhino's functional form (Eq. (6)) proposed in Section 3.1

$$\log p_\theta(X_t^{i,(n)}|\mathbf{Pa}_G^i(<t), \mathbf{Pa}_G^i(t)) = \log p_{g_i}\left(z_t^{i,(n)}|\mathbf{Pa}_G^i(<t)\right) \tag{13}$$

where $z_t^{i,(n)} = X_t^{i,(n)} - f_i(\mathbf{Pa}_G^i(<t), \mathbf{Pa}_G^i(t))$ and $p_{g_i}$ is defined in Eq. (8) (Appendix A for details). The parameters $\theta$, $\phi$ are learned by maximizing the ELBO, where the Gumbel-softmax gradient

estimator is used (Jang et al., 2016; Maddison et al., 2016). We also leverage augmented Lagrangian training (Hestenes, 1969; Andreani et al., 2008), similar as Geffner et al. (2022), to anneal $\alpha, \rho$ to make sure Rhino only produces DAGs (refer to Appendix B.1 in Geffner et al. (2022)). Once Rhino is trained, the temporal causal graph can be inferred by $\boldsymbol{G} \sim q_\phi(\boldsymbol{G})$.

**Treatment effect estimation**    As Rhino learns the causal graph and the functional relationship simultaneously, it can be extended for causal inference tasks such as treatment effect estimation (Geffner et al., 2022). See Appendix D for details.

## 4    THEORETICAL CONSIDERATIONS

Here, we show the theoretical guarantees of Rhino including (1) the structural identifiability of Rhino SEMs and (2) soundness of the proposed variational inference framework. Together, they guarantee the validity of Rhino as a causal discovery method. In the end, we clarify relations to existing works.

### 4.1    STRUCTURAL IDENTIFIABILITY

One of the key challenges for causal discovery with a flexible functional relationship is to show the structural identifiability. Namely, we cannot find two different graphs that induce the same joint likelihood from the proposed functional causal model. In the following, we present a theorem for Rhino SEMs that summarizes our main theoretical contribution.

**Theorem 1** (Identifiability of Rhino SEMs)**.** Assuming Rhino SEMs satisfy the *causal Markov property, minimality, sufficiency, DAGness and the induced joint likelihood has a proper density* (see Appendix B for details), and (1) all functions and induced distributions are third-order differentiable; (2) function $f_i$ is *non-linear* and *not invertible* w.r.t. any nodes in $\mathbf{Pa}_G^i(t)$; (3) the double derivative $(\log p_{g_i}(g_i(\epsilon_t^i)|\mathbf{Pa}_G^i(<t)))''$ w.r.t $\epsilon_t^i$ is zero at most at some discrete points, then the SEM with the form defined in Eq. (6) is structural identifiable for *both bivariate and multivariate case*.

*Sketch of proof.* This theorem is a summary of a collection of theorems proved in Appendix B. The strategy is instead of directly proving the identifiability of the model, we provide identifiability conditions for a general temporal SEMs, followed by showing a generalization of Rhino SEMs satisfies these conditions. The identifiability of Rhino SEMs directly follows from it.

**Prove bivariate identifiability conditions for general temporal SEMs**    The first step is to prove the bivariate identifiability conditions that a general temporal SEMs (Eq. (2)) should satisfy (refer to Theorem 3 in Appendix B.1). Inspired by the techniques from Peters et al. (2013), we proved that temporal SEMs are bivariate identifiable if (1) the model for initial conditions is identifiable; (2) the model is **identifiable w.r.t. instantaneous parents**. Compared to Peters et al. (2013), we relaxed the identifiable model class condition so that it can be applied to history-dependent noise. In particular, (2) implies we only need to pay attention to instantaneous parents, rather than the entire parents (Peters et al., 2013), and opens the door for flexible lagged dependency.

**Identifiability of history-dependent post non-linear model**    Next, we propose a novel generalization of Rhino SEMs, called *history-dependent PNL*. Theorem 4 and Corollary 4.1 in Appendix B.2 prove it is bivariate identifiable w.r.t. instantaneous parents (i.e. satisfy the conditions in Theorem 3) with additional assumptions (1), (2) and (3) in Theorem 1. The functional form of history-dependent PNL is defined as

$$X_t^i = \nu_{it}\left(f_{it}\left(\mathbf{Pa}_G^i(<t), \mathbf{Pa}_G^i(t)\right) + g_{it}\left(\mathbf{Pa}_G^i(<t), \epsilon_{it}\right), \mathbf{Pa}_G^i(<t)\right),$$

where $\nu$ is invertible w.r.t. the first argument. Due to its similarity to PNL (Zhang & Hyvarinen, 2012) and ANM (Hoyer et al., 2008), we combine their techniques to prove our results. The bivariate identifiability of Rhino SEMs directly follows from this with $\nu$ being the identity mapping.

**Generalization to multivariate case**    In the end, we prove the above bivariate identifiability can be generalized to the multivariate case by adapting the techniques from Peters et al. (2012) and combining it with the proof strategy from step 1. Refer to Theorem 5 in Appendix B.3 for details.

$\square$

## 4.2 VALIDITY OF VARIATIONAL OBJECTIVE AND RELATIONS TO OTHER METHODS

Next, we show the validity of the variational objective (Eq. (11)) in the sense that optimizing it can lead to the ground truth graph. Theorem 1 in Geffner et al. (2022) justifies the validity of the variational objective under the same set of assumptions as Rhino.

**Theorem 2** (Validity of variational objective (Geffner et al., 2022))**.** Assuming the conditions in Theorem 1 are satisfied, and further assume no model misspecification (see Definition B.1), then the solution $(\theta', q'_\phi(\boldsymbol{G}))$ from optimizing Eq. (11) with infinite data satisfies $q'_\phi(\boldsymbol{G}) = \delta(\boldsymbol{G} = \boldsymbol{G}')$, where $\boldsymbol{G}'$ is a unique graph. In particular, $\boldsymbol{G}' = \boldsymbol{G}^*$ and $p_{\theta'}(\boldsymbol{X}_{0:T}; \boldsymbol{G}') = p(\boldsymbol{X}_{0:T}; \boldsymbol{G}^*)$, where $\boldsymbol{G}^*$ is the ground truth graph and $p(\boldsymbol{X}_{0:T}; \boldsymbol{G}^*)$ is the true data generating distribution.

We emphasize that Theorem 2 guarantees the correctness with the global optimum, rather than characterizing the convergence during optimizing Eq. (11). We assume the Rhino only searches in the DAG space. In practice, with augmented Lagrangian (Wei et al., 2020; Ng et al., 2022), we can only obtain an approximate posterior over DAGs, converging to a local optimum.

**Relation to other methods**  Many previous works of using functional causal model for causal time series discovery (Hyvärinen et al., 2010; Pamfil et al., 2020; Tank et al., 2018; Peters et al., 2013) are closely related to Rhino. Since Rhino SEMs incorporate history-dependent noise with flexible function non-linearity, it is the most flexible member of this family. Refer to Appendix C for details.

## 5 RELATED WORK

Assaad et al. (2022) provides a comprehensive overview of causal discovery method for time series. The first is Granger causality, which can be further split into (1) vector auto-regression (Wu et al., 2020; Shojaie & Michailidis, 2010; Siggiridou & Kugiumtzis, 2015; Amornbunchornvej et al., 2019) and (2) deep learning (Löwe et al., 2022; Tank et al., 2018; Bussmann et al., 2021; Dang et al., 2018; Xu et al., 2019). Granger causality methods cannot handle instantaneous effects, which can be observed in a slow-sampling system. Additionally, they also assume a fixed noise distribution.

Using funcitonal causal models can mitigate the aforementioned two problems. VARLiNGaM (Hyvärinen et al., 2010) extends the identifiability theory of linear non-Gaussian ANM (Shimizu et al., 2006) to vector auto-regression for modelling time series data. DYNOTEARS (Pamfil et al., 2020) leverages the recently proposed NOTEARS framework (Zheng et al., 2018) to continuously relax the DAG constraints for fully differentiable DAG structure learning. However, the above approach is still limited to linear functional forms. TiMINo (Peters et al., 2013) provides a general theoretical framework for temporal causal discovery with functional causal models. Our theory leverages some of their proof techniques. Unfortunately, all the aforementioned methods assume no history dependency for the noise. On the other hand, Rhino can model (1) non-linear function relations; (2) instantaneous effect; (3) and history-dependent noise at the same time.

The third category is constraint-based approaches. Due to its non-parametric nature, it can handle history-dependent noise. PCMCI (Runge et al., 2019) combines PC (Spirtes et al., 2000) and conditional independence test for discovery from time series. PCMCI$^+$ (Runge, 2018; 2020) further extends PCMCI to infer both lagged and instantaneous effects. CD-NOD (Huang et al., 2020) has recently been proposed to handle non-stationary heterogeneous data, where the data distribution can shift across time. Despite their generality, they can only infer MECs; cannot learn the explicit functional forms between variables; and require a stronger assumption than minimality (i.e. faithfulness).

## 6 EXPERIMENTS

We release the code of Rhino for reproducing the following experiments.[2].

### 6.1 SYNTHETIC DATA

We evaluate our method on a large set of synthetically generated datasets with known causal graphs. We use the main body of this paper to present the overall performance of our method compared to rel-

---

[2]https://github.com/microsoft/causica/tree/v0.0.0

evant baselines and one ablation study on the robustness to lag mismatch. In Appendix F.3, we conduct extensive analysis, including (1) on different graph type; (2) ablation on history-dependency; (3) ablation study on instantaneous effect. This set of datasets are generated by various settings (e.g. type of graphs, instantaneous/no instantaneous effect, etc.). 5 datasets are generated for each combination of settings with different seeds, yielding 160 datasets in total. In order to comprehensively test Rhino's robustness, we deliberately generated 75% of the datasets that mismatch the Rhino configurations. Details of the data generation can be found in Appendix F.1.

We compare Rhino to a wide range of baselines, including VARLiNGaM (Hyvärinen et al., 2010), PCMCI$^+$(Runge, 2020) and DYNOTEARS (Pamfil et al., 2020). PCMCI only outputs Markov equivalence classes (MECs). We resolve this by enumerating all DAGs in the MEC. For details on the methods, see Appendix F.2. Additionally, we include two variants of Rhino: (1) Rhino+g, where an independent Gaussian noise is used; (2) Rhino+s, where Gaussian $\epsilon_i$ is transformed by an independent spline.

Figure 1 presents the $F_1$ score of the lagged, instantaneous and temporal adjacency matrix of all methods aggregated over all datasets[3], denoted as 'Lag', 'Inst.' and 'Temporal', respectively. Rhino achieves overall competitive or the best performance in terms of the full temporal adjacency matrix across all possible datasets, especially for lower dimensions. Comparing Rhino's lagged discovery to its two variants, the better score indicates the history-dependent noise is useful to the lagged graph discovery, contributing to the better overall $F_1$ performance (Appendix F.3 for ablation).

Despite of the strong performance from PCMCI$^+$, it can only identify the graph up to MECs without explicit functional relations. Computationally, PCMCI$^+$ exceeds the maximum training time of **1 week** on 40 nodes (see Appendix F.3), suggesting its computation bottleneck in high dimensions.

DYNOTEARS achieves inferior results in general due to limited modelling power from the linear nature. This is much clearer in high dimensions due to the increasing difficulty of the problem.

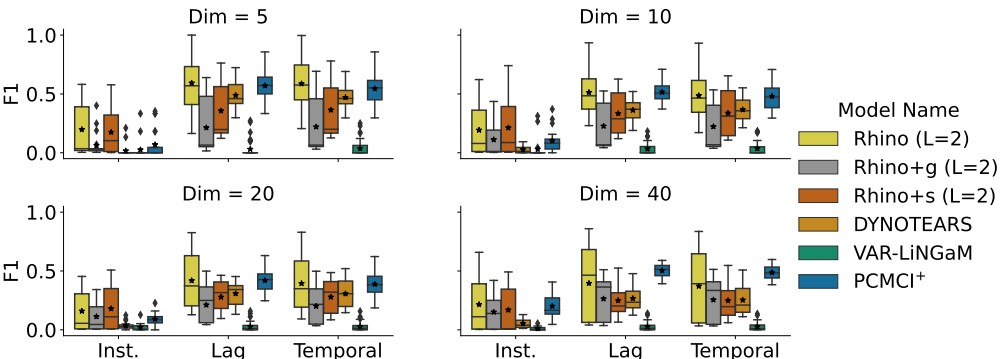

Figure 1: $F_1$-scores of Rhino (light yellow) compared to all baseline methods. The different subplots show the performance for dataset with different number of nodes. 'L=2' refers to models with lag 2.

We explore the behaviour of Rhino with different lag parameters other than the ground truth lag 2. From Table 1, worse training log-likelihoods suggest that Rhino with insufficient history ($\text{lag} = 1$) is unable to correctly model the data and this leads to a decrease in $F_1$ scores. Interestingly, Rhino is also robust with longer lags. Despite of the slightly better likelihood ($\text{lag} = 3$), it achieves comparable performance to the model with the correct lag. Also, from their similar $F_1$ Lag score, it suggests the extra adjacency matrix is nearly empty.

## 6.2 DREAM3 GENE NETWORK

In this section, we evaluate Rhino performance with a real-world biology benchmark called *DREAM3* (Prill et al., 2010; Marbach et al., 2009). These datasets are often used to evaluate Granger causality (Khanna & Tan, 2019; Tank et al., 2018; Nauta et al., 2019; Bussmann et al., 2021) but

---

[3]We note that we run each method on 40 different dataset settings for all possible numbers of nodes.

| Dim | | Rhino (L=1) | Rhino (L=2) | Rhino (L=3) |
|---|---|---|---|---|
| 5 | $F_1$ Inst. | $0.11 \pm 0.17$ | $0.20 \pm 0.22$ | $0.21 \pm 0.23$ |
| | $F_1$ Lag | $0.28 \pm 0.13$ | $0.59 \pm 0.22$ | $0.57 \pm 0.24$ |
| | $F_1$ Temporal | $0.34 \pm 0.12$ | $0.59 \pm 0.20$ | $0.56 \pm 0.22$ |
| | LL | $-4.14 \pm 1.63$ | $-3.83 \pm 1.62$ | $-3.75 \pm 1.64$ |
| 10 | $F_1$ Inst. | $0.13 \pm 0.17$ | $0.19 \pm 0.23$ | $0.19 \pm 0.22$ |
| | $F_1$ Lag | $0.26 \pm 0.08$ | $0.51 \pm 0.17$ | $0.48 \pm 0.19$ |
| | $F_1$ Temporal | $0.28 \pm 0.11$ | $0.49 \pm 0.18$ | $0.45 \pm 0.20$ |
| | LL | $-7.97 \pm 2.09$ | $-7.21 \pm 2.22$ | $-7.01 \pm 1.91$ |
| 20 | $F_1$ Inst. | $0.15 \pm 0.16$ | $0.16 \pm 0.17$ | $0.18 \pm 0.19$ |
| | $F_1$ Lag | $0.24 \pm 0.12$ | $0.42 \pm 0.22$ | $0.40 \pm 0.21$ |
| | $F_1$ Temporal | $0.25 \pm 0.13$ | $0.39 \pm 0.22$ | $0.37 \pm 0.21$ |
| | LL | $-15.62 \pm 3.16$ | $-14.70 \pm 2.87$ | $-14.72 \pm 2.82$ |
| 40 | $F_1$ Inst. | $0.13 \pm 0.16$ | $0.22 \pm 0.23$ | $0.18 \pm 0.21$ |
| | $F_1$ Lag | $0.20 \pm 0.18$ | $0.40 \pm 0.31$ | $0.34 \pm 0.30$ |
| | $F_1$ Temporal | $0.20 \pm 0.18$ | $0.37 \pm 0.30$ | $0.32 \pm 0.29$ |
| | LL | $-31.44 \pm 5.16$ | $-30.10 \pm 4.71$ | $-30.20 \pm 4.74$ |

Table 1: Comparison of the causal discovery performance of Rhino with different lag-parameters ($L \in [1, 3]$). Apart from the 3 $F_1$ scores, LL shows the log-likelihood of the training data.

recently adopted for SEM-based method (Pamfil et al., 2020). The dataset consists *in silico* measurements of gene expression levels for 5 different networks. Each network contains $d = 100$ genes. Each time series represents a perturbation trajectory with time length $T = 21$. For each network, 46 perturbation trajectories are recorded. The goal is to infer the causal structure of each network. We use the *area under the ROC curve* (AUROC) as the performance metric. We consider the same baselines as in the synthetic experiments (i.e. DYNOTEARS and PCMCI$^+$) without *VARLiNGaM* since its default implementation fails when the number of variables ($d = 100$) is greater than the series length ($T = 21$). Additionally, we also consider relevant Granger causality methods, including *cMLP*, *cLSTM* (Tank et al., 2018); *TCDF*(Nauta et al., 2019); *SRU* and *eSRU* (Khanna & Tan, 2019)). Their corresponding results are directly cited from Khanna & Tan (2019). Appendix G.1 specifies Rhino hyperparameters. Since the ground truth graph is a summary graph (see Definition G.1 in Appendix G.2), Appendix G.2 details about the post-processing step on aggregating temporal graph to summary graph for Rhino, DYNOTEARS and PCMCI$^+$.

| Method | E.Coli 1 | E.Coli 2 | Yeast 1 | Yeast 2 | Yeast 3 |
|---|---|---|---|---|---|
| cMLP | 0.644 | 0.568 | 0.585 | 0.506 | 0.528 |
| cLSTM | 0.629 | 0.609 | 0.579 | 0.519 | 0.555 |
| TCDF | 0.614 | 0.647 | 0.581 | 0.556 | 0.557 |
| SRU | 0.657 | 0.666 | 0.617 | 0.575 | 0.55 |
| eSRU | 0.66 | 0.629 | 0.627 | 0.557 | 0.55 |
| DYNO. | 0.590 | 0.547 | 0.527 | 0.526 | 0.510 |
| PCMCI$^+$ | $0.530 \pm 0.002$ | $0.519 \pm 0.002$ | $0.530 \pm 0.003$ | $0.510 \pm 0.001$ | $0.512 \pm 0$ |
| Rhino+g | **0.673±0.013** | 0.665±0.009 | **0.659±0.005** | **0.598±0.004** | **0.588±0.005** |
| Rhino | **0.685±0.003** | **0.680±0.007** | **0.664±0.006** | 0.585±0.004 | 0.567±0.003 |

Table 2: The AUROC of the aggregated adjacency matrix for 5 DREAM3 datasets without self-connections. DYNO. means DYNOTEARS. For Rhino and PCMCI$^+$, the results are reported by averaging over 5 runs. Khanna & Tan (2019) only reported the single-run results for baselines.

Table 2 demonstrates the AUROC of the summary graph inferred after training. It is clear that Rhino and its variant outperform all other methods. Although Rhino is not formulated to solve the summary graph discovery, it shows a clear advantage compared to the state-of-the-art Granger causality. Thus, Rhino can be used to infer either temporal or summary graph depending on users' needs.

By inspecting the hyperparameters of Rhino in Appendix G.1, instantaneous effects seem to provide no obvious help for in these datasets. It suggests the recording intervals are fast enough to avoid any aggregation effect. This explains why the Granger causality can also perform reasonably well.

Unlike the strong performances of DYNOTEARS and PCMCI$^+$ in synthetic experiments, they perform poorly in DREAM3. The linear nature of DYNOTEARS seems to harm its performance drastically. PCMCI$^+$ suffers from the low independence test power under small training data.

Another interesting ablation is to compare with Rhino+g, which performs on par with Rhino and achieves better scores on 2 out of 5 datasets. Although we have no access to the true noise mechanism, we suspect that the added noise is not history-dependent and highly likely to be Gaussian. Despite the model mismatch, Rhino is still one of the best methods for this problem. This further strengthens our belief in the robustness of our model under different setups.

### 6.3 NETSIM BRAIN CONNECTIVITY

In this section, we evaluate Rhino using fMRI imaging data, which has also been used as a benchmark for temporal causal discovery (Löwe et al., 2022; Khanna & Tan, 2019; Assaad et al., 2022). Each time series represents the signal simulated for a human subject, which describes $d = 15$ different regions in the brain with $T = 200$ timestamps. The goal is to infer the connectivity between different brain regions. We assume that different human subjects share the same connectivity. We only use the data from human subject $2 - 6$ in *Sim-3.mat* from `https://www.fmrib.ox.ac.uk/datasets/netsim/index.html` with self-connections. We use the same set of baselines as DREAM3 (Section 6.2) plus VARLiNGaM. Appendix G.4 describes hyperparameter settings.

Table 3 shows the AUROCs for different methods. Remarkably, the proposed Rhino and its variants achieve significantly better AUROC compared to the baselines. Especially, Rhino obtains nearly optimal AUROC, demonstrating its robustness to the small dataset and good balances between true and false positive rates (see Appendix H). By comparing Rhino and Rhino+NoInst., we conclude that modelling instantaneous effects is important in real application, indicating the sampling interval is not frequent enough to explain everything as lagged effects. This can be double confirmed by comparing Rhino+NoInst with Granger causality, where it performs on par with the state-of-the-art baseline when disabling the instantaneous effect. Last but not least, by comparing Rhino+g with Rhino, we find that history-dependent noise is also helpful in this dataset.

| Method | AUROC |
|---|---|
| cMLP | 0.93 |
| cLSTM | 0.83 |
| TCDF | 0.91 |
| SRU | 0.80 |
| eSRU | 0.88 |
| DYNO. | 0.90 |
| PCMCI$^+$ | $0.83 \pm 0$ |
| VARLiNGaM | $0.84 \pm 0$ |
| Rhino+g | $0.974 \pm 0.002$ |
| Rhino+NoInst. | $0.93 \pm 0.006$ |
| Rhino | $\mathbf{0.99 \pm 0.001}$ |

Table 3: AUROCs of the summary graph. Rhino+NoInst is Rhino without instantaneous effects. For Rhino, VARLiNGaM, PCMCI$^+$, results are obtained by averaging over 5 different runs.

## 7 CONCLUSION

Inferring temporal causal graphs from observational time series is an important task in many scientific fields. Especially, some applications (e.g. education, climate science, etc.) require the modelling of non-linear relationships; instantaneous effects and history-dependent noise distributions at the same time. Previous works fail to offer an appropriate solution for all three requirements. Motivated by this, we propose Rhino, which combines vector auto-regression with deep learning and variational inference to perform causal temporal relationship learning with all three requirements. Theoretically, we prove the structural identifiability of Rhino with flexible history-dependent noise, and clarify its relations to existing works. Empirical evaluations demonstrate its superior performance and robustness when Rhino is misspecified, and the advantages of history-dependent noise mechanisms. This opens an exciting route of extending Rhino to handle non-stationary time-series and unobserved confounders in future work.

## 8 REPRODUCIBILITY STATEMENT

**Theoretical Contributions** The main theoretical contribution is summarized in Theorem 1. This theorem is the result from a collection of theorems proved in Appendix B. In Appendix B, we detailed the fundamental assumptions (Assumption 1-Assumption 5) required for the all theorems. The theorem-specific assumptions are mentioned in the statement of the theorem. To ease the understanding of the proof, we also provide the skecth of proof in Theorem 1. Since Theorem 2 is directly cited from (Geffner et al., 2022) without major modification, the proof can be found in Appendix A in Geffner et al. (2022).

**Empirical Evaluations** For synthetic, DREAM3 and Netsim experiments, we listed the hyperparameters in Appendix F.2, Appendix G.1 and Appendix G.4, respectively. Appendix F.1 explains the synthetic data generation. For DREAM3 and Netsim, the dataset can be found in the public github repo `https://github.com/sakhanna/SRU_for_GCI/tree/master/data`. The post processing steps for DREAM3 and Netsim evaluations are described in Appendix G.2.

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

## A ELBO AND LIKELIHOOD DERIVATION

The goal is to derive a lower bound for the joint likelihood $p_\theta(\boldsymbol{X}_{0:T}^{(1)}, \ldots, \boldsymbol{X}_{0:T}^{(N)})$.

$$p_\theta(\boldsymbol{X}_{0:T}^{(1)}, \ldots, \boldsymbol{X}_{0:T}^{(N)})$$

$$= \log \int p_\theta\left(\boldsymbol{X}_{0:T}^{(1)}, \ldots, \boldsymbol{X}_{0:T}^{(N)} | \boldsymbol{G}\right) p(\boldsymbol{G}) d\boldsymbol{G}$$

$$= \log \int \frac{q_\phi(\boldsymbol{G})}{q_\phi(\boldsymbol{G})} p_\theta\left(\boldsymbol{X}_{0:T}^{(1)}, \ldots, \boldsymbol{X}_{0:T}^{(N)} | \boldsymbol{G}\right) p(\boldsymbol{G}) d\boldsymbol{G}$$

$$\geq \int q_\phi(\boldsymbol{G}) \log p_\theta\left(\boldsymbol{X}_{0:T}^{(1)}, \ldots, \boldsymbol{X}_{0:T}^{(N)} | \boldsymbol{G}\right) p(\boldsymbol{G}) d\boldsymbol{G} + H(q_\phi(\boldsymbol{G})) \tag{14}$$

$$= \mathbb{E}_{q_\phi(\boldsymbol{G})}\left[\sum_{n=1}^{N} \log p_\theta(\boldsymbol{X}_{0:T}^{(n)} | \boldsymbol{G}) + \log p(\boldsymbol{G})\right] + H(q_\phi(\boldsymbol{G}))$$

where Eq. (14) is obtained by using Jensen's inequality.

We can further simplify the likelihood $p_\theta(\boldsymbol{X}_{0:T}^{(n)} | \boldsymbol{G})$:

$$\log p_\theta(\boldsymbol{X}_{0:T}^{(n)} | \boldsymbol{G}) = \log \prod_{t=0}^{T} p_\theta(\boldsymbol{X}_t^{(n)} | \boldsymbol{X}_{<t}^{(n)}, \boldsymbol{G})$$

$$= \sum_{t=0}^{T} \log p_\theta\left(\boldsymbol{X}_t^{(n)} | \boldsymbol{X}_{<t}^{(n)}, \boldsymbol{G}\right)$$

$$= \sum_{t=0}^{T} \sum_{i=1}^{D} \log p_\theta\left(X_t^{i,(n)} | \mathbf{Pa}_G^i(<t), \mathbf{Pa}_G^i(t)\right) \tag{15}$$

where Eq. (15) is obtained through Markov factorization (Lauritzen, 1996).

## B STRUCTURAL IDENTIFIABILITY

In this section, we will focus on proving the structural identifiability of Rhino SEMs. Before diving into the details, let us clarify the required assumptions.

**Definition B.1** (Correctly specified model). For a true data generating mechanism $X^i = f_i^*(\mathrm{Pa}^i, \epsilon^i)$ for $i = 1, \ldots, D$, we say a model with functional space $\mathcal{F}$ is correctly specified if there exists a function $f_i \in \mathcal{F}$ s.t. $f_i = f_i^*$ for all $i = 1, \ldots, D$.

Here, we emphasize that the above definition of model specification does not require identifiability in parameters space, but in function space instead. Namely, it allows multiple sets of parameters that correspond to the same function. Our definition is more general than some of the previous work, which enforces parameter identifiability (e.g. Def 2.1 in Ma & Zhang (2021))

**Assumption 1** (Causal Stationarity (Runge, 2018)). The time series process $\boldsymbol{X}_t$ with a graph $\boldsymbol{G}$ is called *causally stationary* over a time index set $\mathcal{T}$ if and only if for all links $X_{t-\tau}^i \to X_t^j$ in the graph

$$X_{t-\tau}^i \not\perp\!\!\!\perp X_t^j | \boldsymbol{X}_{<t} \backslash \{X_{t-\tau}^i\} \text{ holds for all } t \in \mathcal{T}$$

This characterizes the nature of the time-series data generating mechanism, which validates the choice of the auto-regressive model.

**Assumption 2** (Causal Markov Property (Peters et al., 2017)). Given a DAG $\boldsymbol{G}$ and a joint distribution $p$, this distribution is said to satisfy causal Markov property w.r.t. the DAG $\boldsymbol{G}$ if each variable is independent of its non-descendants given its parents.

This is a common assumptions for the distribution induced by an SEM. With this assumption, one can deduce conditional independence between variables from the graph.

**Assumption 3** (Causal Minimality). Consider a distribution $p$ and a DAG $G$, we say this distribution satisfies causal minimality w.r.t. $G$ if it is Markovian w.r.t. $G$ but not to any proper subgraph of $G$.

Minimality is also a common assumption for SEMs (Hoyer et al., 2008; Zhang & Hyvarinen, 2012; Peters et al., 2012), which can be regarded as a weaker version of *faithfulness* (Peters et al., 2017).

**Assumption 4** (Causal Sufficiency). A set of observed variables $V$ is causally sufficient for a process $X_t$ if and only if in the process every common cause of any two or more variables in $V$ is in $V$ or has the same value for all units in the population.

This assumption implies there are no latent confounders present in the time-series data.

**Assumption 5** (Well-defined Density). We assume the joint likelihood induced by the Rhino SEM (Eq. (6)) is absolutely continuous w.r.t. a Lebesgue or counting measure and $|\log p(X_{0:T}; G)| < \infty$ for all possible $G$.

This assumption is to make sure the induced distribution has a well-defined probability density function. It is also required for the equivalence of the global, local Markov property and Markov factorization property (Theorem 6.22 from Peters et al. (2017)).

**Assumption 6** (Rhino in DAG space). We assume the Rhino framework can only return the solutions from DAG space. Namely, the posterior distribution from Rhino can only put weights on DAGs.

This assumption regularizes the search space of Rhino to be DAG space, which aligns with assumption 2 and 3 on causal graphs.

In the following, we will structure the entire proof into three steps:

1. Prove a general conditions that the *bivariate* time series model needs to satisfy for structural identifiability. This adapts from the theorem 1 in Peters et al. (2013).

2. Prove that a generalized form of SEM, modified from the *post non-linear* (*PNL*) model (Zhang & Hyvarinen, 2012), satisfies the conditions mentioned in step 1. The proposed Rhino (Eq. (6)) is a special case of the above SEM.

3. In the end, we generalize the above indentifiability to the *multivariate* case.

### B.1  GENERAL IDENTIFIABILITY CONDITIONS

First, we derive the conditions required for identifiability for a general bivariate time series SEM, defined as

$$X_t^i = f_{i,t}\left(\mathbf{Pa}_G^i(< t), \mathbf{Pa}_G^i(t), \epsilon_t^i\right). \tag{16}$$

We call the above SEM *transition model*, since it only defines the transition behavior rather than the initial conditions. We also need to incorporate a *source model*, which characterizes the initial conditions:

$$X_s^i = f_{i,s}(\mathbf{Pa}_G^i, \epsilon_s^i) \tag{17}$$

for $s \in [0, \mathcal{S}]$, where $\mathcal{S}$ is the length for the initial conditions and $\mathbf{Pa}_G^i$ contains the parents for node $i$. We define $p_s(X_{0:\mathcal{S}})$ as the induced joint distribution for the initial conditions.

Now, we prove the following theorem.

**Theorem 3** (Identifiability conditions for bivariate time series). Assuming Assumption 1-5 are satisfied, given a bivariate temporal process $X_{0:T}$ and $Y_{0:T}$ that are governed by the above SEM (Eq. (16)) with source model (Eq. (17)), then the above SEM for the bivariate temporal process is structural identifiable if the following conditions are true:

1. Source model $f_{i,s}$ is structural identifiable for all $i = 1, \dots, D$ and $s \in [0, \mathcal{S}]$.

2. The transition model (Eq. (16)) is *bivariate identifiable* w.r.t the *instantaneous parents*. Namely, if graph $G$ induced conditional distributions $p(X_t, Y_t | \mathbf{Pa}_G^{X,Y}(< t))$, then $\nexists G'$ such that $G \neq G'$ and the induced conditional $\bar{p}(X_t, Y_t | \overline{\mathbf{Pa}}_{G'}^{X,Y}(< t)) = p$ for all $t \in [\mathcal{S} + 1, T]$.

where $\mathbf{Pa}_G^{X,Y}(<t)$ is the union of the lagged parents of $X_t$ and $Y_t$ under $G$, and $\overline{\mathbf{Pa}}_{G'}^{X,Y}(<t)$ is the union of parents under $G'$.

*Proof.* We prove this by contradiction. Assume we have an induced joint distribution $p(\boldsymbol{X}_{0:T}, \boldsymbol{Y}_{0:T})$ under $G$, and corresponding $\bar{p}$ under $G'$. We further assume the above two conditions in the theorem are met and $p = \bar{p}$ but $G \neq G'$.

Thus, we have $D_{\mathrm{KL}}[p\|\bar{p}] = 0$. Due to the temporal nature of the model, we can further decompose it as the following:

$$D_{\mathrm{KL}}[p\|\bar{p}]$$

$$= \int p(\boldsymbol{X}_{0:T}, \boldsymbol{Y}_{0:T}) \log \frac{p(\boldsymbol{X}_{0:T}, \boldsymbol{Y}_{0:T})}{\bar{p}(\boldsymbol{X}_{0:T}, \boldsymbol{Y}_{0:T})} d\boldsymbol{X}_{0:T} d\boldsymbol{Y}_{0:T}$$

$$= D_{\mathrm{KL}}[\underbrace{p(\boldsymbol{X}_{0:\mathcal{S}}, \boldsymbol{Y}_{0:\mathcal{S}})}_{p_s} \| \underbrace{\bar{p}(\boldsymbol{X}_{0:\mathcal{S}}, \boldsymbol{Y}_{0:\mathcal{S}})}_{\bar{p}_s}] + \int p(\boldsymbol{X}_{0:\mathcal{S}}, \boldsymbol{Y}_{0:\mathcal{S}}) D_{\mathrm{KL}}[p(\boldsymbol{X}_{\mathcal{S}+1:T}, \boldsymbol{Y}_{\mathcal{S}+1:T}|\boldsymbol{X}_{0:\mathcal{S}}, \boldsymbol{Y}_{0:\mathcal{S}})\|$$

$$\bar{p}(\boldsymbol{X}_{\mathcal{S}+1:T}, \boldsymbol{Y}_{\mathcal{S}+1:T}|\boldsymbol{X}_{0:\mathcal{S}}, \boldsymbol{Y}_{0:\mathcal{S}})]d\boldsymbol{X}_{0:\mathcal{S}} d\boldsymbol{Y}_{0:\mathcal{S}}$$

$$= D_{\mathrm{KL}}[p_s\|\bar{p}_s] + \sum_{t=\mathcal{S}+1}^{T} \mathbb{E}_{p(\boldsymbol{X}_{0:t-1}, \boldsymbol{Y}_{0:t-1})} [D_{\mathrm{KL}}[p(X_t, Y_t|\boldsymbol{X}_{0:t-1}, \boldsymbol{Y}_{0:t-1})\|\bar{p}(X_t, Y_t|\boldsymbol{X}_{0:t-1}, \boldsymbol{Y}_{0:t-1})]]$$

$$= 0.$$

This means we have $D_{\mathrm{KL}}[p_s\|\bar{p}_s] = 0$ and $D_{\mathrm{KL}}[p(X_t, Y_t|\boldsymbol{X}_{0:t-1}, \boldsymbol{Y}_{0:t-1})\|\bar{p}(X_t, Y_t|\boldsymbol{X}_{0:t-1}, \boldsymbol{Y}_{0:t-1})] = 0$ almost everywhere. Inspired by the strategy used in (Peters et al., 2013), We consider the following three scenarios:

**Disagree on initial conditions** We assume $G$ and $G'$ disagree on the initial conditions. From the condition 1, we know the source model $f_{i,s}$ is identifiable. Namely, we cannot find $G \neq G'$ with disagreement on initial conditions such that $D_{\mathrm{KL}}[p_s\|\bar{p}_s] = 0$. This is a contradiction, meaning that $G$ and $G'$ must agree on the connections between initial set of nodes.

**Disagree on lagged parents only** This means for all $t \in [\mathcal{S}+1, T]$, the instantaneous connections at $t$ for $G$ and $G'$ are the same, and $\exists t \in [\mathcal{S}+1, T]$ such that $\mathbf{Pa}_G^{X,Y}(<t) \neq \overline{\mathbf{Pa}}_{G'}^{X,Y}(<t)$. We can use a similar argument as the theorem 1 in Peters et al. (2013). W.l.o.g., we assume under $G$, we have $X_{t-\tau} \to Y_t$ and there is no connections between them under $G'$. Thus, from Markov conditions, we have

$$Y_t \perp\!\!\!\perp X_{t-\tau} | \boldsymbol{X}_{0:t-1} \cup \boldsymbol{Y}_{0:t-1} \cup \mathrm{ND}_t^Y \backslash \{Y_t, X_{t-\tau}\}$$

under $G'$, where $\mathrm{ND}_t^Y$ are the non-descendants of node $Y_t$ at some time $t$. However, from the causal minimality and proposition 6.16 in Peters et al. (2017), we have

$$Y_t \not\perp\!\!\!\perp X_{t-\tau} | \boldsymbol{X}_{0:t-1} \cup \boldsymbol{Y}_{0:t-1} \cup \mathrm{ND}_t^Y \backslash \{Y_t, X_{t-\tau}\}$$

under $G$. This means under this case, $D_{\mathrm{KL}}[p(X_t, Y_t|\boldsymbol{X}_{0:t-1}, \boldsymbol{Y}_{0:t-1})\|\bar{p}(X_t, Y_t|\boldsymbol{X}_{0:t-1}, \boldsymbol{Y}_{0:t-1})] \neq 0$, which is a contradiction.

**Disagree also on instantaneous parents** This scenarior means $\exists t \in [\mathcal{S}+1, T]$ such that they disagree on instantaneous parents. W.l.o.g. we assume $X_t \to Y_t$ under $G$ and $Y_t \to X_t$ under $G'$.

Let's define $\boldsymbol{X}_{0:t-1} \cup \boldsymbol{Y}_{0:t-1} = \boldsymbol{h}$, $\boldsymbol{h}_G^Y \subseteq \boldsymbol{h}$ contains the values of $\mathbf{Pa}_G^Y(<t)$ under $G$, $\bar{\boldsymbol{h}}_{G'}^Y \subseteq \boldsymbol{h}$ contains the parent values under $G'$, and $\boldsymbol{h}_G^X$, $\bar{\boldsymbol{h}}_{G'}^X$ accordingly. Thus, the induced conditional distributions from SEM (Eq. (16)) with $G$, $G'$ are

$$p(X_t, Y_t|\boldsymbol{h}_G^X \cup \boldsymbol{h}_G^Y) \quad \text{and} \quad \bar{p}(X_t, Y_t|\bar{\boldsymbol{h}}_{G'}^X \cup \bar{\boldsymbol{h}}_{G'}^Y)$$

From the Markov conditions, we have

$$p(X_t, Y_t|\boldsymbol{X}_{0:t-1}, \boldsymbol{Y}_{0:t-1}) = p(X_t, Y_t|\mathbf{Pa}_G^{X,Y}(<t))$$

Therefore, we have

$$D_{\mathrm{KL}}\left[p(X_t, Y_t|\boldsymbol{h})\|\bar{p}(X_t, Y_t|\boldsymbol{h})\right]$$
$$=0$$
$$=D_{\mathrm{KL}}[p(X_t, Y_t|\boldsymbol{h}_G^X \cup \boldsymbol{h}_G^Y)\|\bar{p}(X_t, Y_t|\bar{\boldsymbol{h}}_{G'}^X \cup \bar{\boldsymbol{h}}_{G'}^Y)]$$

for arbitrary $\boldsymbol{h}$, which contradicts the structural identifiability w.r.t. the instantaneous parents.

In summary, with the two conditions, we cannot find $\boldsymbol{G} \neq \boldsymbol{G}'$ such that the induced joint $p(\boldsymbol{X}_{0:T}, \boldsymbol{Y}_{0:T}) = \bar{p}(\boldsymbol{X}_{0:T}, \boldsymbol{Y}_{0:T})$, meaning that the SEMs defined as Eq. (16) and Eq. (17) are identifiable w.r.t. *bivariate time series*. □

Since one can use any identifiable static models to characterize the initial behavior of the time series, we will focus on condition 2 for the transition model. In the following, we will show that a generalization of PNL, called *history-dependent PNL*, satisfies condition 2 under assumptions.

### B.2 IDENTIFIABILITY OF HISTORY-DEPENDENT PNL

First, we propose a generalization of PNL (Zhang & Hyvarinen, 2012) so that it can be history-dependent. For a multivariate temporal process $\boldsymbol{X}_{0:T}$, we propose *history-dependent PNL* as

$$X_t^i = \nu_{it}\left(f_{it}\left(\mathbf{Pa}_G^i(<t), \mathbf{Pa}_G^i(t)\right) + g_{it}\left(\mathbf{Pa}_G^i(<t), \epsilon_{it}\right), \mathbf{Pa}_G^i(<t)\right) \tag{18}$$

where $\nu_{it}$ is an invertible transformation w.r.t. the first argument. The main differences of the above SEM compared to typical PNL are (1) the invertible transformation $\nu_{it}$ can be history dependent; (2) the inner noise distribution can also be history-dependent.

Next, we show the main theorem about its bivariate identifiability w.r.t. its instantaneous parents.

**Theorem 4** (History-dependent PNL Bivariate Identifiability). Assume Assumption 1-5 are satisfied, all transformations in Eq. (18) and corresponding induced distributions are $3^{rd}$-order differentiable. Given a bivariate temporal process $\boldsymbol{X}_{0:T}, \boldsymbol{Y}_{0:T}$, then the history-dependent PNL defined as Eq. (18) is bivariate identifiable w.r.t its instantaneous parents (i.e. satisfy condition 2 in Theorem 3), except for some special cases.

*Proof.* W.l.o.g. at time $t \in [\mathcal{S}+1, T]$, we assume $X_t \to Y_t$ for instantaneous connection under $\boldsymbol{G}$ and $Y_t \to X_t$ under $\boldsymbol{G}'$. We fix a value $\boldsymbol{h}$ for their entire history $\boldsymbol{X}_{0:t-1} \cup \boldsymbol{Y}_{0:t-1} = \boldsymbol{h}$. With $\boldsymbol{h}$, we further define their lagged parents as $\mathbf{Pa}_G^X(<t) = \boldsymbol{h}_G^X \subseteq \boldsymbol{h}$, $\mathbf{Pa}_G^Y(<t) = \boldsymbol{h}_G^Y \subseteq \boldsymbol{h}$ under $\boldsymbol{G}$ and $\overline{\mathbf{Pa}}_{G'}^X(<t) = \bar{\boldsymbol{h}}_{G'}^X \subseteq \boldsymbol{h}$, $\overline{\mathbf{Pa}}_{G'}^Y(<t) = \bar{\boldsymbol{h}}_{G'}^Y$ under $\boldsymbol{G}'$.

Therefore, the SEM at time $t$ can be written as

$$Y_t = \nu\left(f(\boldsymbol{h}_G^Y, X_t) + g(\boldsymbol{h}_G^Y, \epsilon_Y), \boldsymbol{h}_G^Y\right) \tag{19}$$

and

$$X_t = \bar{\nu}\left(\bar{f}(\bar{\boldsymbol{h}}_{G'}^X, Y_t) + \bar{g}(\bar{\boldsymbol{h}}_{G'}^X, \epsilon_X), \bar{\boldsymbol{h}}_{G'}^X\right) \tag{20}$$

under $\boldsymbol{G}$ and $\boldsymbol{G}'$, respectively. Let's assume that their induced conditional distributions at time $t$ are equal (i.e. violating the identifiable condition (2) in Theorem 3):

$$\underbrace{\log p(X_t, Y_t|\boldsymbol{h}_G^X \cup \boldsymbol{h}_G^Y)}_{\text{under } \boldsymbol{G}} = \underbrace{\log \bar{p}(X_t, Y_t|\bar{\boldsymbol{h}}_{G'}^X \cup \bar{\boldsymbol{h}}_{G'}^Y)}_{\text{under } \boldsymbol{G}'}$$

From the Markov properties, the above equation is equivalent to

$$\log p(X_t, Y_t|\boldsymbol{h}) = \log \bar{p}(X_t, Y_t|\boldsymbol{h})$$

with a fixed value $\boldsymbol{h}$ of the entire history.

Now, let's define

$$\alpha_t = \bar{\nu}^{-1}(X_t) \quad \text{and} \quad \beta_t = \nu^{-1}(Y_t)$$

where we omits the dependence of $\bar{\nu}^{-1}$ to $\bar{h}_{G'}^X$ and $\nu^{-1}$ to $h_G^Y$. It is easy to observe that we have an invertible mapping between $(X_t, Y_t)$ and $(\alpha_t, \beta_t)$. Thus, from the change of variable formula, we have

$$\log p(X_t, Y_t | \boldsymbol{h}) = \log p_{\alpha,\beta}(\alpha_t, \beta_t | \boldsymbol{h}) + \log |\boldsymbol{J}|$$

and

$$\log \bar{p}(X_t, Y_t | \boldsymbol{h}) = \log \bar{p}_{\alpha,\beta}(\alpha_t, \beta_t | \boldsymbol{h}) + \log |\boldsymbol{J}|$$

where $\boldsymbol{J}$ is the Jacobian matrix of the transformation. Thus, the equivalence of $\log p$ and $\log \bar{p}$ in the $(X_t, Y_t)$ space can be translated to $(\alpha_t, \beta_t)$ space.

Thus, from Eq. (19), we have

$$\beta_t = \Phi(\alpha_t) + g(\boldsymbol{h}_G^Y, \epsilon_Y) \tag{21}$$

under $\boldsymbol{G}$. And from Eq. (20), we have

$$\alpha_t = \Psi(\beta_t) + \bar{g}(\bar{\boldsymbol{h}}_{G'}^X, \epsilon_X) \tag{22}$$

under $\boldsymbol{G'}$. This forms an *additive noise model* between $\alpha_t, \beta_t$ with history-dependent noise. Next, we can use a similar proof techniques as in Hoyer et al. (2008). Here, $\Phi(\cdot) = f(\boldsymbol{h}_G^Y, \cdot) \circ \bar{\nu}(\bar{\boldsymbol{h}}_{G'}^X, \cdot)$ and $\Psi(\cdot) = \bar{f}(\bar{\boldsymbol{h}}_{G'}^X, \cdot) \circ \nu(\boldsymbol{h}_G^Y, \cdot)$. We further define

$$\eta_1(\alpha_t) = \log p(\alpha_t | \boldsymbol{h}) \qquad\qquad \bar{\eta}_1(\beta_t) = \log \bar{p}(\beta_t | \boldsymbol{h})$$
$$\eta_2(g(\boldsymbol{h}_G^Y, \epsilon_Y)) = \log p_g(g(\boldsymbol{h}_G^Y, \epsilon_Y)|\boldsymbol{h}) \qquad \bar{\eta}_2(\bar{g}(\bar{\boldsymbol{h}}_{G'}^X, \epsilon_X)) = \log \bar{p}_g(\bar{g}(\bar{\boldsymbol{h}}_{G'}^X, \epsilon_X)|\boldsymbol{h})$$

Thus, under $\boldsymbol{G}$ (i.e. Eq. (21)), we have

$$\log p(\alpha_t, \beta_t | \boldsymbol{h}) = \log p(\beta_t | \alpha_t, \boldsymbol{h}) + \log p(\alpha_t | \boldsymbol{h})$$
$$= \eta_2(\beta_t - \Phi(\alpha_t)) + \eta_1(\alpha_t) \tag{23}$$

Similarly, under $\boldsymbol{G'}$ (i.e. Eq. (22)), we have

$$\log \bar{p}(\alpha_t, \beta_t) = \bar{\eta}_2(\alpha_t - \Psi(\beta_t)) + \bar{\eta}_1(\beta_t) \tag{24}$$

Based on Eq. (24), we have

$$\frac{\partial^2 \log \bar{p}}{\partial \alpha_t \partial \beta_t} = -\bar{\eta}_2'' \Psi' \quad \text{and} \quad \frac{\partial^2 \log \bar{p}}{\partial \alpha_t^2} = \bar{\eta}_2''$$

Thus, we have

$$\frac{\partial}{\partial \alpha_t}\left(\frac{\partial^2 \log \bar{p}/\partial \alpha_t \partial \beta_t}{\partial^2 \log \bar{p}/\partial \alpha_t^2}\right) = 0$$

Due to the equivalence of $\log \bar{p}$ and $\log p$, we apply the above operations to Eq. (23). After some algebraic manipulation, we obtained the following differential equations for $\eta_2'' \Phi' \neq 0$:

$$\eta_1''' - \frac{\eta_1'' \Phi''}{\Phi'} = \left(\frac{\eta_2' \eta_2'''}{\eta_2''} - 2\eta_2''\right)\Phi'' \Phi' - \frac{\eta_2'''}{\eta_2''}\Phi' \eta_1'' + \eta_2'\left(\Phi''' - \frac{(\Phi'')^2}{\Phi'}\right). \tag{25}$$

Interestingly, this is exactly equivalent to Eq.(4) in Zhang & Hyvarinen (2012). The main difference is the definition of variables and transformations in here are all history-dependent.

Further, we can also observe that

$$\beta_t \perp\!\!\!\perp \bar{g}(\boldsymbol{h}_G^Y, \epsilon_Y)|\boldsymbol{X}_{0:t-1} \cup \boldsymbol{Y}_{0:t-1} = \boldsymbol{h}.$$

Since $\beta_t = \Phi(\alpha_t) + g(\boldsymbol{h}_G^Y, \epsilon_Y)$ and $\bar{g}(\bar{\boldsymbol{h}}_{G'}^X, \epsilon_X) = \alpha_t - \Psi(\beta_t)$, it is trivial to show the determinant of the Jacobian of the transformation $(\alpha_t, g)$ to $(\beta_t, \bar{g})$ is 1. Thus, by a similar argument in theorem 1 from Zhang & Hyvarinen (2012), we can derive

$$\frac{1}{\Psi'} = \frac{\eta_1'' + \eta_2''(\Phi')^2 - \eta_2'\Phi''}{\eta_2''\Phi'}$$

for $\eta_2''\Phi' \neq 0$.

Thus, the above two differential equations has the same form as theorem 1 in Zhang & Hyvarinen (2012) where the main difference is that all distributions and transformations involved in our case depend on history $\boldsymbol{h}$.

Therefore, we can directly cite the theorem 8 from Zhang & Hyvarinen (2012), which proves that the above differential equations hold true only for 5 types of special cases. One can refer to Table 1 in Zhang & Hyvarinen (2012) for details. $\qquad\square$

Corollary 10 from Zhang & Hyvarinen (2012) validates the choice of using nueral network for the transformation $f$. For completeness, we include it here with slight modification:

**Corollary 4.1** (Identifiability with neural netowrk $f$). Assuming the assumptions in Theorem 4 are true, and the double derivative $(\log p_g(g(\mathbf{Pa}_G^Y(<t), \epsilon_Y) | \boldsymbol{X}_{0:t-1} \cup \boldsymbol{Y}_{0:t-1}))''$ w.r.t $\epsilon_Y$ is zero at most at some discrete points. If function $f$ is not invertible *w.r.t. the instantaneous parents*, then, the history-dependent PNL defined as Eq. (18) is *bivariate identifiable w.r.t. the instantaneous parents* (i.e. satisfy condition 2 in Theorem 3).

It is clear to see that the form of Rhino SEMs (Eq. (6)) is a special case of the history-dependent PNL (Eq. (18)), where the outer history-dependent invertible transformation $\nu$ is the identity mapping. Thus, we can directly leverage Theorem 3 together with Theorem 4 to show Rhino SEMs are identifiable w.r.t bivariate time series, and Corollary 4.1 to validate our design choice (Eq. (7)).

### B.3   GENERALIZING TO MULTIVARIATE TIME SERIES

Previously, we prove the identifiability conditions for bivariate time series. In this section, we will generalize it to the multivariate case.

**Theorem 5** (Generalization to multivariate time series). Assuming the assumptions in Theorem 4 are satisfied, we further assume that the multivariate SEM defined in Eq. (18) satisfies: for each pair of node $i, j \in \boldsymbol{V}$, the SEM

$$X_t^i = \nu_{it}\left(f_{it}\left(\mathbf{Pa}_G^i(<t), \mathbf{Pa}_G^i(t)\backslash\{X_t^j\}, \underbrace{\cdot}_{X_t^j}\right) + g_{it}\left(\mathbf{Pa}_G^i(<t), \epsilon_{it}\right), \mathbf{Pa}_G^i(<t)\right)$$

is *bivariate identifiable* w.r.t. the input, and an identifiable source model is adopted. Then, the history-dependent PNL is *identifiable except for some special cases*.

*Proof.* For this proof, we can follow the strategy used in Theorem 3 and Peters et al. (2013). We categorize the difference of the graph $G$ and $G'$ into three types. Following the same analysis of the $KL$ divergence of the two induced joint distributions, we can see that (1) $D_{\text{KL}}[p_s\|\bar{p}_s] = 0$ and $D_{\text{KL}}[p(\boldsymbol{X}_t|\boldsymbol{X}_{0:t-1})\|\bar{p}(\boldsymbol{X}_t|\boldsymbol{X}_{0:t-1})] = 0$.

**Disagree on initial conditions**   Since we assume that the source model is identifiable, this contradicts $D_{\text{KL}}[p_s\|\bar{p}_s] = 0$.

**Disagree on lagged parents only**   We notice that the analysis used in Theorem 3 for this disagreement can be directly translated to multivariate case. The only difference is that the notation $Y_t$, $X_t$ is changed accordingly.

**Disagree also on instantaneous parents**   For this case, with a fixed history value $\boldsymbol{h} = \boldsymbol{X}_{0:t-1}$, the aim is to compare the conditionals $D_{\text{KL}}[p(\boldsymbol{X}_t|\boldsymbol{X}_{0:t-1} = \boldsymbol{h})\|\bar{p}(\boldsymbol{X}_t|\boldsymbol{X}_{0:t-1} = \boldsymbol{h})]$. Thus, the problem becomes to how to generalize the bivariate identifiability for instantaneous parents to the multivariate case. We leverage the theorem 2 from Peters et al. (2012), which proves the multivariate identifiability for any models that belongs to IFMOC. It is easy to see that if the assumptions in Theorem 5 are met, the history-dependent PNL belongs to IFMOC *w.r.t. the instantaneous parents*. It should be noted that the entire history-dependent PNL *DOES NOT belong to IFMOC*, but this does not affect our results since we only care about the instantaneous parents under this case.   □

## C   RELATION TO OTHER METHODS

**VARLiNGaM (Hyvärinen et al., 2010)**   VARLiNGaM (Hyvärinen et al., 2010) is a causal discovery method for time series data based on the linear vector auto-regression, which can model both lagged and instantaneous effects. Its SEM is defined as Eq. (5), where the noise $\epsilon_t^i$ is an independent non-Gaussian noise. It is easy to observe that this is a special case of Rhino (Eq. (6)) by setting $f_i$ as the matrix multiplication of the weighted adjacency $\boldsymbol{G}_{0:K}$ with the nodes, and $g_i$ as the identity mapping. For the training objective, VARLiNGaM adopted a two stage training to sidestep the

difficulty of directly optimizing the log likelihood. From the Theorem 2 for Rhino, we note that the solution from optimizing the variational objective is equivalent to maximizing the log likelihood under infinite data limit. Therefore, by setting large enough DAGness penalty coefficient $\alpha$, $\rho$, the inferred graph from both methods should be equivalent.

**DYNOTEARS (Pamfil et al., 2020)** The formulation of DYNOTEARS is the same as VAR-LiNGaM, which is based on linear vector auto-regression. The main novelty is the usage of the DAGness penalty $h(\boldsymbol{G})$, which continuously relaxes the DAG constraint. The training objective is the mean square error with augmented Lagrange scheme for DAGness penalty. Thus, it is obvious that DYNOTEARS is a special case of Rhino with linear transformations and identity $g_i$. Similarly, Theorem 2 shows the connections between the variational objective and maximum likelihood, which is equivalent to mean square error if the noise distribution is *Gaussian with equal variances*.

**cMLP** cMLP (Tank et al., 2018) combines Granger causality with deep neural networks. The model formulation is
$$X_t^i = f_i(\boldsymbol{X}_{0:t-1}^1, \ldots, \boldsymbol{X}_{0:t-1}^D) + \epsilon_t^i$$
where $f_i$ is a function based on MLP. Although the input is the entire history, the one that matters is the node that has the connection to $X_t^i$ (i.e. lagged parents). Therefore, it is easy to see they are closely related to Rhino without *instantaneous parents* $\mathbf{Pa}_G^i(t)$ and history-dependent noise. Since the training objective of cMLP is based on the mean square error with sparseness constraint, by the same argument as before, the variational objective is equivalent to mean square error with equal variance Gaussian noise and large training data.

**TiMINo (Peters et al., 2013)** TiMINo is most similar to our work among all the aforementioned methods in terms of model formulation. TiMINo proposed a very general formulation based on IF-MOC (Peters et al., 2012) and showed the conditions for structural identifiability. Rhino generalizes the TiMINO in a way such that noise history dependency can be incorporated. Thus, Rhino only belongs to IFMOC w.r.t. the instantaneous parents. Therefore, Rhino without the history-dependent noise is a TiMINo model. The training objective of TiMINo is based on the dependence minimization between the noise residuals and causes, and can only infer summary graph instead of temporal causal graph. Zhang et al. (2015) proved the equivalence of the mutual information minimization to maximum likelihood, which is equivalent to our variational objective under infinite data.

## D    TREATMENT EFFECT ESTIMATION

We now show how to leverage the fitted Rhino for estimating the *conditional average treatment effect* (CATE). For simplicity, we only consider a special case of CATE defined as

$$\text{CATE}(a, b) = \mathbb{E}_{q_\phi(\boldsymbol{G})} \left[ \mathbb{E}_{p(\boldsymbol{X}_{t+\tau}^Y | \boldsymbol{X}_{<t}, \text{do}(X_t^I = a), \boldsymbol{G})}[X_{t+\tau}^Y] - \mathbb{E}_{p(\boldsymbol{X}_{t+\tau}^Y | \boldsymbol{X}_{<t}, \text{do}(X_t^I = b), \boldsymbol{G})}[X_{t+\tau}^Y] \right]$$
(26)

We assume the conditioning variable can only be $\boldsymbol{X}_{<t}$ (i.e. the entire history before $t$), and the intervention and target variable can only be either at current time $t$ or sometime in the future $t+\tau$. We emphasize that this formulation is for simplicity, and Rhino can be easily generalized to more cases as Geffner et al. (2022). Once fitted, the idea is to draw target samples $X_{t+\tau}^Y$ from the interventional distribution $p(\boldsymbol{X}_{t+\tau}^Y | \boldsymbol{X}_{<t}, \text{do}(X_t^I), \boldsymbol{G})$ for each graph sample $\boldsymbol{G} \sim q_\phi(\boldsymbol{G})$. Then, unbiased Monte Carlo estimation can be used to compute CATE. For sampling from the interventional distribution, we can use the "multilated" graph $\boldsymbol{G}_{\text{do}(X_t^I)}$ to replace $\boldsymbol{G}$, where all incoming edges to $X_t^I$ are removed. The intervention samples can be obtained by simulating the Rhino with history $\boldsymbol{X}_{<t}$, $X_t^I = a$ or $b$ and $\boldsymbol{G}_{\text{do}(X_t^I)}$.

### D.1    CAUSAL INFERENCE RESULTS

Here, we provide the preliminary results for CATE performance of Rhino by calculating the RMSEs of the estimated CATEs comparing to the true CATE from the interventional samples (lower is better). We present boxplots of the performance in Fig. 2. All Rhino-based method perform similarly. Surprisingly, the CATE performance seems to have little correlation to the causal discovery performance and warrants further study in the future.

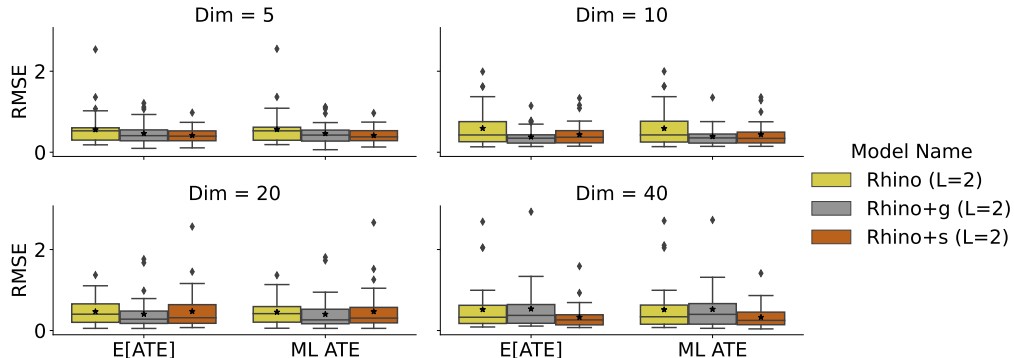

Figure 2: Comparison of the RMSE of the average treatment effects (CATEs) of the different instantiations of Rhino depending on the dimensionality. $\mathbb{E}[\text{CATE}]$ refers to RMSE of the expected CATE over the posterior graph distribution (i.e. $\boldsymbol{G} \sim q_\phi(\boldsymbol{G})$). ML ATE uses the most likely graph to calculate the ATE. These results are obtained by averaging 160 datasets, similar to the discovery setup.

# E  VARIATIONAL DISTRIBUTION FORMULATION

Here we provide the detailed formulation of the independent Bernoulli distribution $q_\phi(\boldsymbol{G})$. Since this distribution is responsible for modelling the temporal adjacency matrix $\boldsymbol{G}_{0:K}$, we use $\Sigma_k$ to represents the edge probability in $\boldsymbol{G}_k$. We further split the edge probability matrices into the instantaneous part $\Sigma_0$ and lagged parts $\Sigma_{1:K}$.

To avoid the constrained optimization of $\Sigma_{1:K}$ (i.e. the value needs to be within $[0, 1]$), we adopt the following formulation:

$$\sigma_{k,ij} = \frac{\exp(u_{k,ij})}{\exp(u_{k,ij}) + \exp(v_{k,ij})} \tag{27}$$

where $u_{k,ij} \in \boldsymbol{U}_k$, $v_{k,ij} \in \boldsymbol{V}_k$ and $\boldsymbol{U}_k, \boldsymbol{V}_k \in \mathbb{R}^{D \times D}$ for all $k = 1, \ldots, K$. Since we do not require lagged adjacency matrix to be a DAG, $\boldsymbol{U}_k, \boldsymbol{V}_k$ has no constraints during optimization.

On the other hand, $\boldsymbol{G}_0$ needs to be a DAG for instantaneous effect. By smart formulation, we can get rid of the length-1 cycles. The intuition is that for a pair of node $i, j$, only three mutually exclusive possibilities can exist: (1) $i \rightarrow j$; (2) $j \rightarrow i$; (3) no edge between them. Thus, instead of using a full probability matrix $\Sigma_0$, we use three lower triangular matrices $\boldsymbol{U}_0$, $\boldsymbol{V}_0$ and $\boldsymbol{E}_0$ to characterise the above three scenarios. For node $i > j$,

$$p(i \rightarrow j) = \frac{\exp(u_{ij})}{\exp(u_{ij}) + \exp(v_{ij}) + \exp(e_{ij})}$$
$$p(j \rightarrow i) = \frac{\exp(v_{ij})}{\exp(u_{ij}) + \exp(v_{ij}) + \exp(e_{ij})}$$
$$p(\text{no edge}) = \frac{\exp(e_{ij})}{\exp(u_{ij}) + \exp(v_{ij}) + \exp(e_{ij})}.$$

Thus, by this formulation, the corresponding instantaneous adjacency matrix will not contain length-1 cycles.

# F   SYNTHETIC EXPERIMENTS

## F.1   DATA GENERATION

We create the synthetic datasets in a four step process: 1) generate random Erdös–Rényi (ER) or scale-free (SF) graphs that specify the lagged and instantaneous causal relationships; 2) drawing random MLPs for the functional relationships as well as a random *conditional* spline transformation to modulate the scale of the Gaussian noise variables $\epsilon$; 3) sample initial starting conditions and follow Eq. (2) with the additive noise to simulate the temporal progression; 4) removing the burn-in period and return stable timeseries. We consider four different axes of variation for the data generation: number of nodes $N_{nodes} \in [5, 10, 20, 40]$; ER or SF graphs; instantaneous or no instantaneous effects; and history-dependent or history-independent noise (i.e. Gaussian noise). All combinations are generated with 5 different seeds, yielding 160 different datasets. Datasets with instantaneous effects have $4 \times N_{nodes}$ edges in the instantaneous adjacency matrix. All datasets have $2 \times N_{nodes}$ connections in the lagged adjacency matrices. The MLPs for the functional relationships are fully-connected with two hidden layers,64 units and ReLU activation. In case of history-independent noise, we are using Gaussian as the base distribution. The history dependency is modelled as a product of a scale variable obtained by the transformation of the averaged lagged parental values through a random-sampled quadratic spline, and Gaussian noise variable.

The datasets with 40 nodes are generated with a series length of 400 steps, a burn-in period of 100 steps, and 100 training series. All other datasets are generated with a time-series length of 200, burn-in period of 50 steps and 50 training series. We generate random interventions for all the datasets by setting the treatment variable to 10 for intervention and -10 for reference. 5000 ground-truth intervention samples are used to estimate the true treatment effect.

## F.2   METHODS

All benchmarks for the synthetic experiments are run by using publicly available libraries: VARLiNGaM Hyvärinen et al. (2010) is implemenented in the `lingam`[4] python package. PCMCI$^+$(Runge, 2020) is implemented in `Tigramite`[5]. We use the implementation in `causalnex`[6] to run DYNOTEARS(Pamfil et al., 2020). We use the default parameters for all these baselines. For PCMCI$^+$, we enumerate all graphs in the Markov equivalence class to evaluate the causal discovery performance (see Appendix G.2 for details).

For Rhino and its variants, we use the same set of hyper-parameters for all 160 datasets to demonstrates our robustness. By default, we allow Rhino and its variants to model instantaneous effect; set the model lag to be the ground truth 2 except for ablation study; the $q_\phi(\boldsymbol{G})$ is initialized to favour sparse graphs (edge probability$< 0.5$); quadratic spline flow is used to for history-dependent noise. For the model formulation, we use 2 layer fully connected MLPs with 64 (5 and 10 nodes), 80 (10 nodes) and 160 (40 nodes) for all neural networks in Rhino-based methods. We also apply layer normalization and residual connections to each layer of the MLPs. For the gradient estimator, we use the Gumbel softmax method with a hard forward pass and a soft backward pass with temperature of 0.25. All spline flows uses 8 bins. The embedding sizes for transformation (i.e. Eq. (7) and conditional spline flow) is equal to the node number.

For the sparseness penalty $\lambda_s$ in Eq. (10), we use 9 for Rhino and Rhino+s, and 5 for Rhino+g. We set $\rho = 1$ and $\alpha = 0$ for all Rhino-based methods. For optimization, we use Adam (Kingma & Ba, 2014) with learning rate 0.01. The training procedure follows from Appendix B.1 in Geffner et al. (2022).

## F.3   ADDITIONAL CAUSAL DISCOVERY RESULTS

**Ablation: different type of graphs**   The first study is to test our model robustness to different types of graphs. Fig. 3 shows the discovery performance over ER or SF graph averaged over all other possible data setting combinations. Most methods perform better on ER graphs than on SF graphs,

---

[4]see `https://lingam.readthedocs.io`
[5]see `https://jakobrunge.github.io/tigramite/`
[6]see `https://causalnex.readthedocs.io/en/latest/`

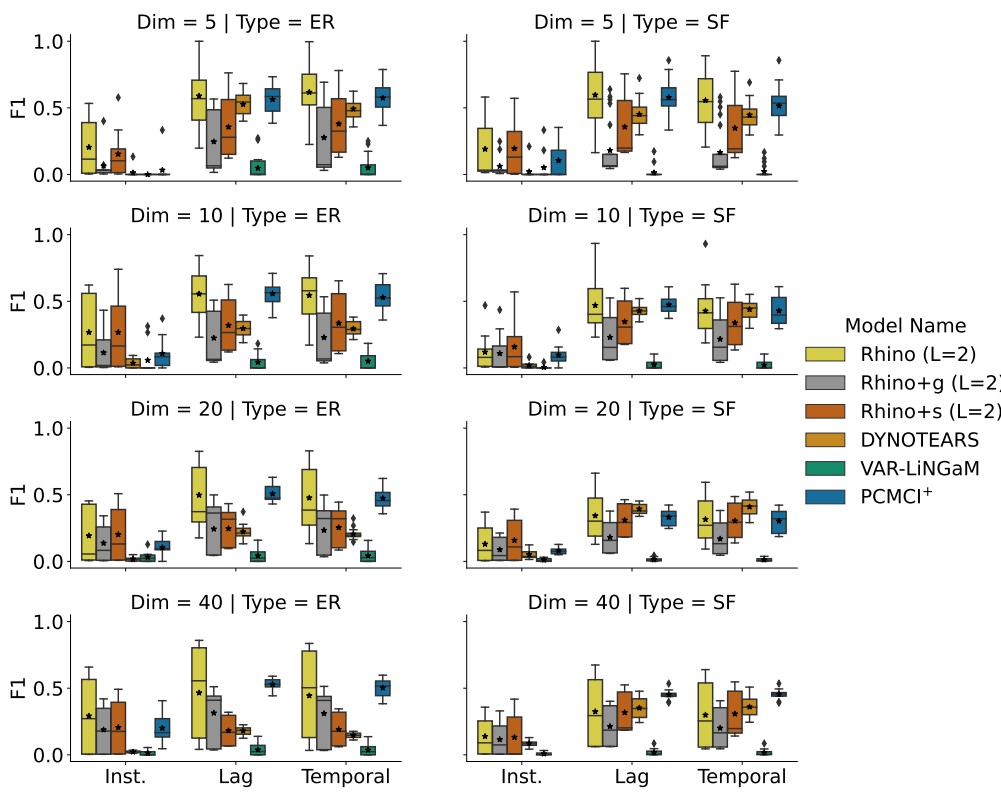

Figure 3: Comparison of the $F_1$ score of the different baseline methods as well as Rhino (light yellow) depending on the dimensionality and the graph type. Inst. refers to the performance on the instantaneous adjacency matrix, Lag refers to the lagged adjancency matrices and temporal considers the full temporal matrix.

with only DYNOTEARS (Pamfil et al., 2020) as an exception. We note that the PCMCI$^+$ runs on SF graphs with 40 nodes exceed our maximum run time of **1 week**, showing its computational limitation in high dimensions. Nevertheless, Rhino achieves consistent performance throughout all graph settings.

**Ablation: history dependency** Figure 4 explores the performance difference of all methods on data generated with/without history-dependent noise. Interestingly, most methods perform better on the history-dependent datasets than the history-independent ones. The possible reasons are (1) the difficulty of the discovery also depends on the randomly sampled functions; (2) the default hyperparameters of all methods are initially chosen to favor the datasets with history-dependent noise and instantaneous effects. We find that PCMCI$^+$ is the most robust across both settings, followed by Rhino and DYNOTEARS. On the other hand, the two variants of Rhino seems to be less robust. When the Rhino is correctly specified, it achieves the best performance. In summary, Rhino demonstrates reasonable robustness to history-dependency mismatch and achieves the best when correctly specified.

**Ablation: instantaneous effect** We investigate the impact of instantaneous effects in the data. Figure 5 shows the $F_1$ score averaged over all possible setting combinations other than instantaneous effect. All methods seem to be robust across both settings with PCMCI$^+$ and Rhino performing the best. The score of the instantaneous adjacency matrix when instantaneous effects are disabled is not defined and therefore not plotted.

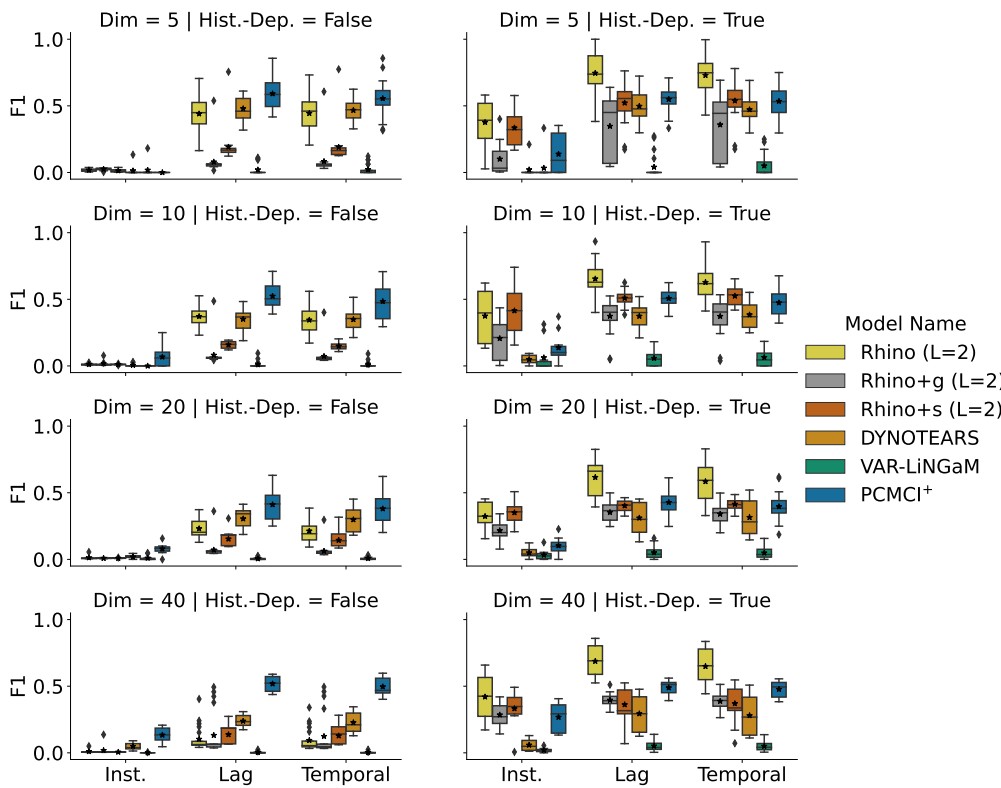

Figure 4: Comparison of the $F_1$ score of the different baseline methods as well as Rhino (light yellow) depending on the dimensionality and whether the data is generated with history-depence or not. Inst. refers to the performance on the instantaneous adjacency matrix, Lag refers to the lagged adjacency matrices and temporal considers the full temporal matrix.

## G  REAL-WORLD EXPERIMENT DETAILS

### G.1  DREAM3 HYPERPARAMETER SETTING

For tuning the hyper-parameters of Rhino, its variants and DYNOTEARS, we split each of the 5 datasets into $80\%/20\%$ training/validation. We tune Rhino and its variants based on the validation likelihoods, and DYNOTEARS based on the validation RMSE error. For PCMCI$^+$, we use the default settings recommended in the Tigramite package (https://github.com/jakobrunge/tigramite). For other Granger causality baselines, refer to Table 7-11 in Khanna & Tan (2019).

Other than the hyper-parameters reported in Table 4, we use 1-layer MLPs with 10 hidden units for both $\ell_{\tau,j}, \zeta_i$ in Eq. (7) and the hyper-network for conditional spline flow (8 bins). All the MLPs use residual connections and layer-norm at every hidden layer. We use linear conditional spline flow (Dolatabadi et al., 2020) instead of the original quadratic version (Durkan et al., 2019) for better training stability. We also initialise the Bernoulli probability $q_\phi(\boldsymbol{G})$ to favour dense graphs (i.e. edge probability $> 0.5$). For prior $p(\boldsymbol{G})$, we set the initial value $\rho = 1$ and $\alpha = 0$. For the gradient estimator, we use the Gumbel softmax method with a hard forward pass and a soft backward pass with temperature of 0.25. We use batch size 64, learning rate 0.001 with Adam optimizer (Kingma & Ba, 2014). The training procedure follows from Appendix B.1 in Geffner et al. (2022).

Table 5 contains the hyper-parameters setup for DYNOTEARS. We set the maximum training iterations to be 1000 with DAGness tolerance $10^{-8}$. The threshold value for the weighted adjacency matrix is 0.05. For PCMCI$^+$, the maximum lag is set to 2. The conditional independence test is set to parcorr, which is based on linear ordinary least square (OLS). A more powerful choice can be

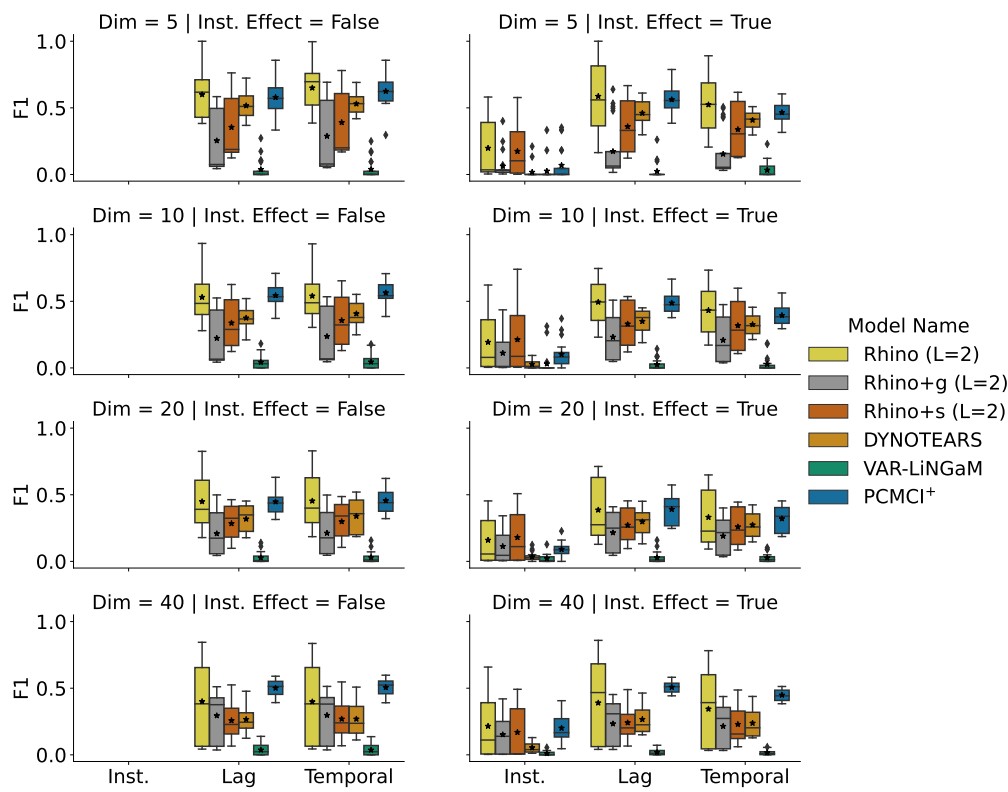

Figure 5: Comparison of the $F_1$ score of the different baseline methods as well as Rhino (light yellow) depending on the dimensionality and whether the data is generated with instantaneous effects or not. Inst. refers to the performance on the instantaneous adjacency matrix, Lag refers to the lagged adjancency matrices and temporal considers the full temporal matrix.

| Hyperparams | Node Embedding | Instantaneous eff. | Node Embed. (flow) | lag | $\lambda_s$ | Auglag |
|---|---|---|---|---|---|---|
| Rhino (Ecoli1) | 16 | False | 16 | 2 | 19 | 30 |
| Rhino (Ecoli2) | 16 | False | 100 | 2 | 25 | 80 |
| Rhino (Yeast1) | 32 | False | 100 | 2 | 25 | 10 |
| Rhino (Yeast2) | 32 | False | 100 | 2 | 25 | 80 |
| Rhino (Yeast3) | 32 | False | 16 | 2 | 25 | 5 |
| Rhino+g (Ecoli1) | 100 | False | N/A | 2 | 15 | 60 |
| Rhino+g (Ecoli2) | 100 | False | N/A | 2 | 25 | 25 |
| Rhino+g (Yeast1) | 100 | False | N/A | 2 | 15 | 5 |
| Rhino+g (Yeast2) | 100 | False | N/A | 2 | 19 | 125 |
| Rhino+g (Yeast3) | 100 | False | N/A | 2 | 9 | 10 |

Table 4: The hyperparameter setup for Rhino. `Node embedding` is the dimensionality of $\boldsymbol{u}_{\tau,i}$ below Eq. (7); `Instantaneous eff.` specifies whether it models the instantaneous effect or not; `Node Embed. (flow)` represents the dimensionality of the node embedding for the hyper-network used for conditional spline flow $g_i$ since the hyper-network shares the similar structure as Eq. (7); `lag` defines the model lag order; and $\lambda_s$ is the sparseness penalty in the prior (Eq. (10)); `Auglag` is the number of augmented Lagrangian steps, each step consists of 2000 training iterations.

a nonlinear independence test based on GP, called `GPDC`. However, PCMCI$^+$ with $GPDC$ is too slow to finish the training.

| Hyperparams | lag | $\lambda_a$ | $\lambda_w$ |
|---|---|---|---|
| Ecoli1 | 2 | 0.01 | 0.5 |
| Ecoli2 | 2 | 0.1 | 0.01 |
| Yeast1 | 2 | 0.005 | 0.1 |
| Yeast2 | 3 | 0.01 | 0.01 |
| Yeast3 | 2 | 0.01 | 0.005 |

Table 5: The hyperparameter setup for DYNOTEARS.

| Metrics | Ecoli1 | Ecoli2 | Yeast1 | Yeast2 | Yeast3 |
|---|---|---|---|---|---|
| Rhino | **0.183± 0.012** | **0.214± 0.009** | **0.261± 0.006** | 0.136± 0.005 | 0.113± 0.005 |
| Rhino+g | 0.169± 0.010 | 0.211±0.010 | **0.254± 0.006** | **0.148± 0.007** | **0.126± 0.003** |
| Dynotears | 0.120 | 0.066 | 0.059 | 0.092 | 0.045 |
| PCMCI | 0.051±0.006 | 0.049± 0.003 | 0.068±0.005 | 0.046± 0.002 | 0.060±0 |

Table 6: Orientation F1 score of DREAM3 datasets.

## G.2 Post-processing temporal adjacency matrix

The ground truth graphs for DREAM3 and Netsim datasets are summary graph, which is essentially the temporal graph aggregated over time. We provide a formal definition of summary graph:

**Definition G.1** (Causal summary graph (Assaad et al., 2022))**.** Let $X_t$ be a multivariate temporal process, and $G = (V, E)$ be a summary graph. The edge $p \rightarrow q$ exists if and only if there exists some time $t$ and some lag $\tau$ such that $X_{t-\tau}^p$ causes $X_t^q$ at time $t$ with a lag $0 \leq i$ for $p \neq q$ and with a time lag of $0 < i$ for $p = q$.

Unlike the some of the Granger causality baselines, Rhino (and its variants), DYNOTEARS, VAR-LiNGaM produces the temporal adjacency matrix after training. For DREAM3 and Netsim datasets, this creates the incompatibility during evaluation. Thus, we need to aggregate the temporal graph into a summary graph before comparing to the ground truth. For binary adjacency matrix, we sum over the time steps followed by a step function, i.e. $\text{step}(\sum_k G_k)$. Thus, there will be an edge $i \rightarrow j$ in summary graph as long as there is a connection from $i$ to $j$ at any timestamp. For the Bernoulli probability matrix from Rhino and its variants, we take a $\max(\cdot)$ over the timestamp to generate the probability matrix for the summary graph.

An exception is PCMCI$^+$, which can only produce MECs for the instantaneous adjacency matrix. In such case, we will enumerate up to 10000 possible instantaneous DAGs from the MECs. Together with the lagged adjacency matrix, we will perform the above post-processing step to generate the corresponding aggregated adjacency matrix. We also estimate the corresponding edge probabilities by taking the average over all possible DAGs.

For DREAM3 experiments, we ignore the self-connections by setting the diagonal of the aggregated adjacency matrix to be 0.

For Netsim, self-connections are not ignored, following the same settings as Khanna & Tan (2019).

## G.3 Additional DREAM3 Results

Here, Fig. 6 shows the additional ROC curve plots for all 5 datasets in DREAM3. For the visualization purpose, we only select a single run for Rhino and this will not affect the curve much due to small standard error in Table 2.

In addition, we provides the additional metrics (Orientation F1 and SHD) for DREAM3 datasets in Table 6 and Table 7. These results are obtained by using the same hyperparameters mentioned in Appendix G.1. In particular, we tune the threshold for rounding the continuous-valued adjacency matrix to binary adjacency matrix for both Rhino and baselines. It can be observed that the F1 and SHD agree with the trend of AUROC, where Rhino and Rhino+g achieve the best performance compared to baselines. This further supports the advantages of our proposed methods.

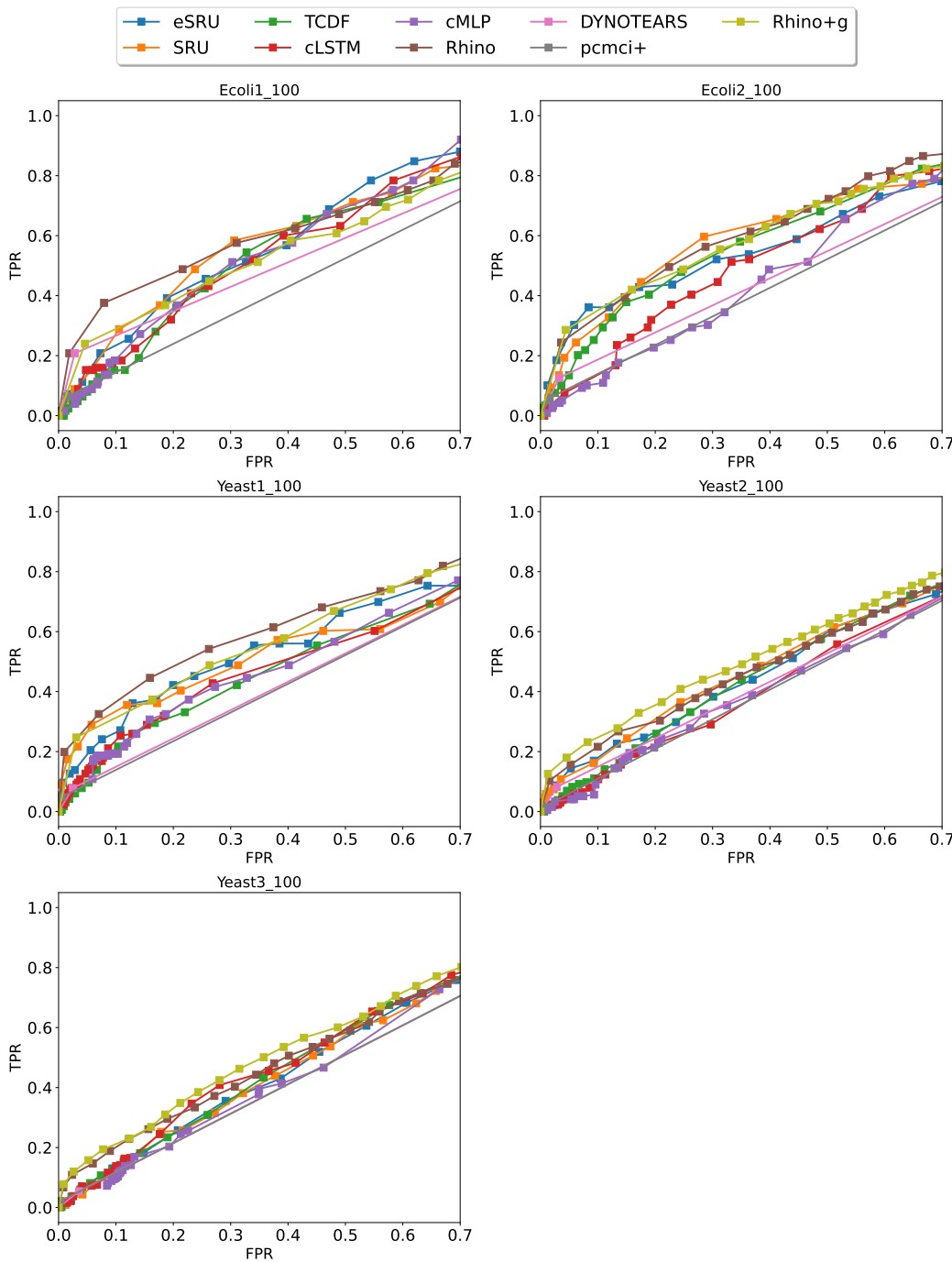

Figure 6: The ROC curve plots of Rhino and other baselines for DREAM3 datasets. For illustration purpose, we only select a single run of Rhino, Rhino+g, DYNOTEARS and PCMCI$^+$ to plot ROC curve. Since the standard error reported in Table 2 is relatively small, the plot should not vary much for other runs. The ROC curve of other baselines are directly taken from figure 7 in Khanna & Tan (2019).

| Metrics | Ecoli1 | Ecoli2 | Yeast1 | Yeast2 | Yeast3 |
|---|---|---|---|---|---|
| Rhino | **157.8±3.23** | **120.4± 1.19** | **161.8±1.86** | **399.8±6.77** | **627.6±5.47** |
| Rhino+g | 162.8±3.13 | **122±2.87** | 182.4±1.43 | **401.4±3.85** | 760.6±6.18 |
| Dynotears | 372 | 422 | 397 | 630 | 838 |
| PCMCI | 266.6±2.01 | 273.2±1.48 | 288.6±2.24 | 507.4±0.92 | 704±0 |

Table 7: SHD of DREAM3 datasets.

## G.4 NETSIM HYPERPARAMETER SETTING

For the Netsim experiment, we extract subject 2-6 in *Sim-3.mat* to form the training data and use subject 7-8 as validation dataset. Following the same settings as DREAM3 (Appendix G.1), we tune the hyperparameters of Rhino and its variants based on the validation log likelihood; DYNOTEARS with MSE on validation dataset; and use default settings of PCMCI$^+$ from Tigramite package.

It is worth noting that unlike DREAM3 experiment, where the results and hyperparameters of Granger causality baselines can be directly taken from Khanna & Tan (2019). Their setup of Netsim experiment is different from ours, where they train the baselines using a **single subject** and compute the corresponding AUROC, followed by averaging over subjects 2-6. Our setup is to train all methods using the entire data from subject 2-6 before computing AUROC. Thus, the hyperparameters for Granger causality are slightly different, and the AUROC increases for the baselines compared to those reported in Khanna & Tan (2019).

**Rhino** The hyperparameters are the same as DREAM3, except for the following: we initialise the Bernoulli probability of $q_\phi(G)$ to have no preference (i.e. edge probability= 0.5); the $\lambda_s = 25$; we use 2 layer MLPs with 64 hidden units for both functional model (Eq. (7)) and hyper-network with embedding size 15; the augmented Lagrangian step is 5. For Rhino variants, we use the above settings as well.

**DYNOTEARS, PCMCI$^+$ and VARLiNGaM** For DYNOTEARS, we set lag to be 2, $\lambda_a = 0.5$ and $\lambda_w = 0.5$. For PCMCI$^+$, we use `parcorr` independence test with lag 3. For VARLiNGaM, we use lag 2 with default settings as `https://lingam.readthedocs.io/en/latest/`.

**Granger Causality** For computing AUROC, we follow the same method as Khanna & Tan (2019); Tank et al. (2018) by sweeping through a range of hyperparameters. Specifically, we use the same hyperparameters for SRU and eSRU as (Khanna & Tan, 2019). For cMLP, we choose the ridge penalty as $0.43$ and sweep through the group sparse penalty in range $[0.1, 1]$. For cLSTM, we set the ridge penalty to be 0.045, and sweep the group sparse penalty in range $[0.1, 1]$.For TCDF, we sweep through the threshold in range $[-1, 2]$ for the attention scores. Other than the above hyperparameters, everything else follows the setup as in Khanna & Tan (2019).

## G.5 ADDITIONAL NETSIM RESULTS

Figure 7 shows the ROC curve plot for Rhino and other baselines. It is clear that Rhino achieves significantly better TPR-FPR trade-offs compared to others. Table 8 shows additional discovery metrics of Rhino and baselines for Netsim dataset. We can observe that F1 score and SHD in general agree with AUROC reported in Table 3, where Rhino-based methods outperform the baselines, apart from Rhino+NoInst. This again confirms the necessity of modelling instantaneous effect for real-world challenges. Rhino outperforms Rhino+g on two out of three metrics (including AUROC), which shows the advantage of history-dependent noise.

# H AUROC METRIC

AUROC metric is a one of the standard metrics for evaluating the causal discovery, which measures the trade-off between the *true positive rate* (TPR) and *false positive rate* (FPR). However, during the experiments, we found out that AUROC does not necessarily correlate well with other discovery metrics. From Fig. 8, it is clear that the $F_1$ score continues to increase whereas AUROC and validation likelihood starts to decrease after few steps. Since the dataset of Netsim is relatively small,

| Method | Ori. F1 | SHD |
|--------|---------|-----|
| DYNO. | 0.341 | 17 |
| PCMCI$^+$ | 0.41 | 18 |
| VarLiNGaM | 0.44 | 18 |
| Rhino+g | $0.539 \pm 0.036$ | $\mathbf{10.4 \pm 1.08}$ |
| Rhino+NoInst | $0.212 \pm 0.014$ | $29 \pm 1.5$ |
| Rhino | $\mathbf{0.551 \pm 0.048}$ | $13.8 \pm 1.5$ |

Table 8: Orientation F1 and SHD of Rhino and baselines for Netsim dataset.

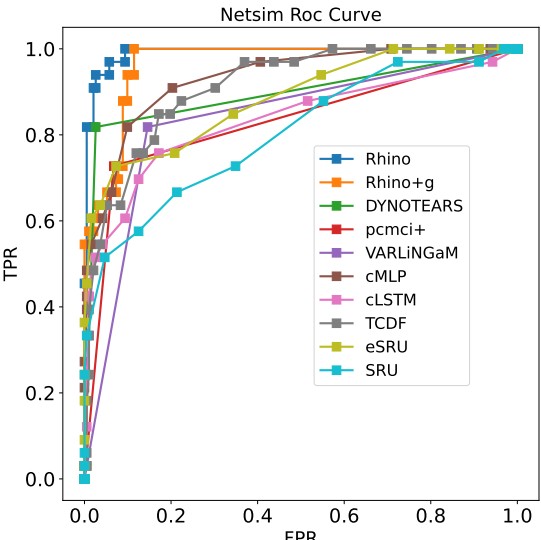

Figure 7: The ROC curve plots of Rhino and other baselines for Netsim dataset. Similar to Fig. 6, we only select 1 run out of 5 for Rhino, Rhino+g, DYNOTEARS, PCMCI$^+$ for illustration purpose.

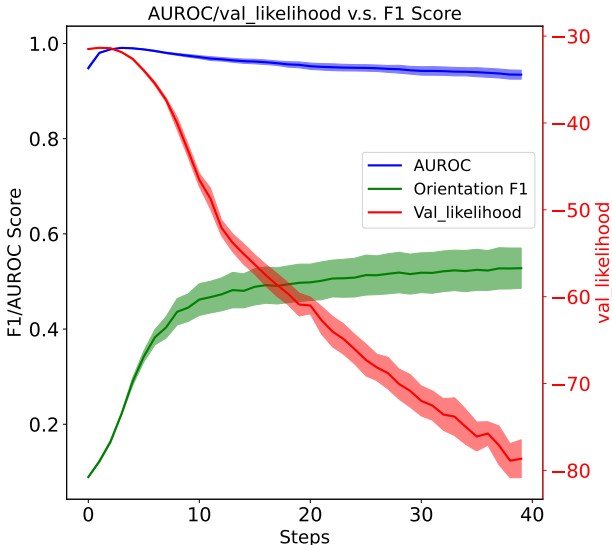

Figure 8: The curves of orientation $F_1$, AUROC and validation likelihood during training. Each curve is obtained by averaging over 5 random seeds. The validation curve agrees well with the AUROC curve, but shows an opposite trends as $F_1$ curve. This potentially indicates model overfitting in the later stage of training.

this indicates the possible overfitting. This disagreement originates from the different aspects these metrics care about. For AUROC, it cares about the trade-off between TPR and FPR with various decision thresholds, and it penalizes the wrong decisions with certainty harshly. On the other hand, $F_1$ score cares about the final inferred binary adjacency matrix with a fixed decision threshold. For example, if we multiply the Bernoulli probability matrix by a small factor (e.g. $10^{-5}$), the AUROC score will remain the same but the $F_1$ score will tends to 0 with the default decision threshold 0.5.

Thus, model overfitting tends to drive the edge probabilities towards 1 or 0, which may help the $F_1$ score but these extreme decisions can result in a large decrease in the AUROC score. Thus, for small dataset, we believe AUROC is a better metric than $F_1$, which also agrees with validation likelihood.

In addition, the Bayesian setup of Rhino may also help with better AUROC for small dataset. From the same figure, even the large decrease of validation likelihood suggests potential model overfitting, the AUROC still maintains a reasonable value. This may be due to the Bayesian view of the causal graph, where the posterior edge probability does not converge to extreme values.

