# OpenReview forum: "Rhino: Deep Causal Temporal Relationship Learning with History-dependent Noise"
_ICLR.cc/2023/Conference — ICLR 2023 notable top 25%_

### Official Review · Reviewer_wW3u · 2022-10-25

**Confidence:** 4
**Clarity, Quality, Novelty And Reproducibility:** This is a well written paper with eno…
**Correctness:** 4
**Technical Novelty And Significance:** 3
**Empirical Novelty And Significance:** 3
**Recommendation:** 8

**Strength And Weaknesses:**

Strength:

1. The motivation of the paper is clearly stated, and the writing of the paper is good. It is easy to follow. Even for the hardest proof part, the sketch of the proof is easy to follow.

2. The experiment part is solid. The experiments are based on datasets from different domains, and all of them show the effectiveness of the proposed model.

3. The mathematic derivation of the paper is solid.

**Summary Of The Paper:**

The author proposed a new structural equation model, named Rhino, that can be used to model causal relations between variables in temporal data. Compared to ANM, the proposed model is more flexible, but there is no sacrificing in identifiable properties. The proposed Rhino model is actually identifiable under some common assumptions. The experiments show that the proposed approach can be applied in various different applications to produce satisfactory results.

**Summary Of The Review:**

The authors proposed a new SEM, named Rhino, for temporal data, and proved the identifiable of the proposed model. Experiments on data from different domains show the effectiveness of the proposed model.

---

> ### Author Response · Authors · 2022-11-10
> **Author's response**
>
> We really appreciate your acknowledgement and your positive opinions about our paper.
>
> We want to ask whether you have any specific points that you are not happy with or have concerns with. Your feedback is valuable for us to further improve the quality of our paper.

---

### Official Review · Reviewer_hiTa · 2022-10-25

**Confidence:** 3
**Correctness:** 3
**Technical Novelty And Significance:** 3
**Empirical Novelty And Significance:** 3
**Recommendation:** 8

**Clarity, Quality, Novelty And Reproducibility:**

The contributions over prior works are clear. The paper is well written. The discussion of related work is adequate. Some part of the clarity can be improved, as stated above.

**Strength And Weaknesses:**

Strengths
- The identifiability result and identification method both seem to be sound.
- Theoretical and empirical results are provided.
- The motivation of using history-dependent noise is convincing as far as I know.

Weaknesses/Main Points
- Several proof techniques are adapted from prior works. It would be clearer to explicitly acknowledge/summarize in the main paper (Section 4.1) if any of the important proof technique are adapted from prior works, instead of only mentioning in the Appendix.
- Some of the writing may be confusing. Is Rhino a type of structural equation model (SEM), or a structural identification method? In Section 3.1 and Theorem 1, it seems that Rhino is defined as a certain type of structural equation model Eq. (5). However, in Section 3.2 and many other parts, it seems that Rhino is a structure identification method that refers to Eq. (10). I think it would be clearer to make a distinction between SEM/identifiability and identification. E.g., we often use LINGAM to refer to the specific structural equation model, and ICA-LINGAM/Direct-LINGAM to refer to the specific method.
- Section 3.2 mentions that augmented Lagrangian training is used to make sure Rhino only produces DAGs. Does Theorem 2 require the augmented Lagrangian training or the DAG assumption of Rhino's solution?
    - If yes:
        - The authors should make this clear in the theorem statement. The current statement seems like any value of $\rho$ and $\alpha$ suffices, and the solution (or search space) need not be DAG.
        - Is there any theoretical guarantee that Rhino only produces DAGs with such training? What assumptions are needed? Such assumptions should also be clearly stated in the theorem statement.
    - If no:
        - Why do we still need such DAG penalty and augmented Lagrangian training?
- What values of $\lambda$ in Eq. (10) is needed for Theorem 2 to hold? For infinite data, do we still need sparseness prior? (In many statistics/ML settings, sparseness is often used for finite samples.)
- Theorem 2: What does it specifically mean to be no model misspeficiation? Could the authors provide a precise technical definition?
- The paper should make it very clear that Theorem 2 is based on global solution of Eq. (10). In practice, it is impossible and we often end up with a local solution.

**Summary Of The Paper:**

This paper provides a new structure identifiability result for temporal causal discovery, specifically time-series structural equation model with nonlinear lagged and instantaneous effects. The key advancement is that the identifiability result allows for history-dependent noise, while existing works typically assume independent noise. The paper further proposes a structure identification method based on continuous optimization, variational inference, and deep learning. Empirical results for synthetic and real world data are further reported.

**Summary Of The Review:**

The setting considered with history-dependent noise is well-motivated. Theoretical results are provided to justify the proposed method. However, there are some concerns with Theorem 2.

---

> ### Author Response · Authors · 2022-11-10
> **Author's response**
>
> We thank the reviewer for the time and provide detailed comments and feedback. Here are our explanations for your concerns.
>
> ### Q1: Explicit mentioning of the proof technique
> In the revised paper, we have explicitly pointed out the relationships between the proof strategies from the previous work and our contributions. I will briefly summarize in the following. The Theorem 1 consists of three proof steps:
> - Step 1: 'Prove bivariate identifiability conditions for general temporal SEMs', are inspired by the proof techniques used in [1]. However, our result is more general in the sense that we do not assume the entire model to be identifiable but only require the identifiability w.r.t. the instantaneous parents, which opens the door for more flexible lagged dependency.
> - Step 2: 'Identifiability of history-dependent post-non-linear model'. Due to the similarity of history-dependent PNL and PNL, we combine the proof techniques used in ANM [2] and PNL [3] to prove the identifiability under conditions.
> - Step 3: 'Generalization to multivariate case'. For this proof, we adapted the strategy used in [4] and combined it with the logic in step 1 to generalize the bivariate identifiability to the multivariate cases.
>
> ### Q2: What does Rhino represent
> Thanks for noticing this ambiguity. Here, Rhino refers to the entire discovery framework, including the Rhino SEMs (Eq.5) and variational training framework (Eq.11). Namely, Rhino is ICA-LiNGaM in your example, and we use "Rhino SEMs" to refer to the class of SEMs, similarly to LiNGaM. In the revised paper, we explicitly distinguish the Rhino SEMs with the variational inference framework. We point out that the identifiability (Theorem 1) is for Rhino SEMs.
>
> ### Q3: Augmented Lagrangian for theorem 2
> From assumptions 2 and 3 in appendix B, which are required by theorem 2 to hold, we have assumed Rhino only works with DAGs $G$. Namely, we assume the solution returned by Rhino is a distribution over DAG space. Although the variational framework alone does not guarantee a DAG solution during optimization, we should emphasize that theorem 2 does not characterize the convergence property of optimizing Eq.11 (i.e. what does the solution look like when the optimization steps $n\rightarrow \infty$). Instead, it answers the question that if the global optimum of Eq.11 has been obtained, the solution must be the ground truth DAG $G^*$.
> In practice, we need to regularize the search space to be a DAG space. Thus, the augmented Lagrangian is to enforce this constraint. We also revised the paper to be clear that the global optimum is required for theorem 2 to hold.
>
> ### Q4: Requirement of sparseness $\lambda$
> Theorem 2 does not need sparseness $\lambda$. Based on the assumptions of Theorem 2, we only have one unique DAG graph (from identifiability and model specification assumption), which corresponds to the global optimum of Eq.11 under infinite data, regardless of the value of $\lambda$. Another way to look at it is since the $\lambda$ only appears in the prior, it plays insignificant roles compared to the likelihood term with infinite data. However, in practice with finite data, the sparsity constraint helps us to discover a sparse graph, which helps to achieve causal minimality.
>
> ### Q5: Precise definition of the correctly specified model
> We have added the precise definition of the correct model specification in the revised paper in appendix B. In a nutshell, we say a model is correctly specified if the model functional space contains the true generating mechanisms. To be precise, for a true data generation mechanism ${X}^i=f_i^*({\textbf{Pa}}^i, \epsilon^i)$ for $i=1,\ldots, D$, we say a model with functional space $\mathcal{F}$ is correctly specified if there exists a function $f_i\in\mathcal{F}$ s.t. $f_i=f_i^*$ for all $i=1,\ldots,D$.
> Here, we should emphasize that we do not require parameter identifiability, and instead only require correct specification in the function space. Namely, we can allow multiple sets of model parameters that correspond to the same function. This is more general than some of the previous work (See Def 2.1 in [5])
>
> ### Reference
> [1] Peters, Jonas, Dominik Janzing, and Bernhard Schölkopf. "Causal inference on time series using restricted structural equation models." Advances in Neural Information Processing Systems 26 (2013).
>
> [2] Hoyer, Patrik, et al. "Nonlinear causal discovery with additive noise models." Advances in neural information processing systems 21 (2008).
>
> [3] Zhang, Kun, and Aapo Hyvarinen. "On the identifiability of the post-nonlinear causal model." arXiv preprint arXiv:1205.2599 (2012).
>
> [4] Peters, Jonas, et al. "Identifiability of causal graphs using functional models." arXiv preprint arXiv:1202.3757 (2012).
>
> [5] Ma, C., & Zhang, C. (2021). Identifiable Generative Models for Missing Not at Random Data Imputation. Advances in Neural Information Processing Systems, 34, 27645-27658.

---

> > ### Comment · Reviewer_hiTa · 2022-11-14
> > **Comments**
> >
> > I appreciate the detailed reply and updated paper from the authors. Many of my concerns have been addressed, and here are some of my further comments (to help make Theorem 2 more rigorous):
> > - As the authors pointed out, the assumptions 2 and 3 listed in Appendix B (and Theorem 1) focus on the ground truth Rhino SEM (i.e. the ground truth graph has to be a DAG). However, I did not manage to understand how it directly relates to "we assume the solution returned by Rhino is a distribution over DAG space". The former is an assumption on the **Rhino SEM**, while the latter is about the **solution of the Rhino method**. Therefore, the latter part is exactly what I think should be further incorporated in the assumption of Theorem 2. Specifically, the current statement of Theorem 2 "the solution from optimizing Eq. (11) with infinite data" does not explicitly mention that "the solution returned by Rhino is a distribution over DAG space".
> > - "Instead, it answers the question that if the global optimum of Eq.11 has been obtained, the solution must be the ground truth DAG G∗. In practice, we need to regularize the search space to be a DAG space.": Similar to my concern above, Eq. 11 (which involves Eq. 10 with the DAG penalty) still does not directly imply that the global optimum of Eq. 11 has to be acyclic. That is, the global optimum of Eq. 11 may not be a DAG, unless the authors explicitly assume it in the theorem statement.
> > - "In practice, we need to regularize the search space to be a DAG space. Thus, the augmented Lagrangian is to enforce this constraint": I think adding this discussion right after Theorem 2 can be very helpful for readers like me. Specifically, it should be emphasized that: (1) the theorem statement, as the authors mentioned, assumes that Rhino method returns an exact DAG, (2) with augmented Lagrangian method, theoretically we are able to get an exact DAG (Wei et al., 2020; Ng et al., 2022), but practically speaking we are only able to get an approximate DAG. Explaining this slight discrepancy between theory and practice can be useful for practitioners.
> >
> > Refs:
> > Wei et al., DAGs with No Fears: A Closer Look at Continuous Optimization for Learning Bayesian Networks, 2020.
> > Ng et al., On the Convergence of Continuous Constrained Optimization for Structure Learning, 2022.

---

> > > ### Author Response · Authors · 2022-11-14
> > > **Author's response 2**
> > >
> > > We really appreciate the feedback and related references from the reviewer. We further edit the paper by
> > > 1. Adding assumption 6 in appendix B to explicitly mention the search space of Rhino is DAG space.
> > > 2. This assumption 6 is mentioned in theorem 1, where theorem 2 also shares the same set of assumptions.
> > > 3. Adding a statement right after theorem 2 to explicitly mention that in practice only local optimum and approximate DAGs can be obtained even with theoretical guarantees.
> > > 4. Adding the given related references.
> > >
> > > Hope this can address the concerns you mentioned above.

---

> > > > ### Comment · Reviewer_hiTa · 2022-11-19
> > > > **Comments**
> > > >
> > > > I appreciate that the authors further incorporated my comments. My concerns have been addressed and I have updated my score.

---

> > > ### Author Response · Authors · 2022-11-17
> > > **End of Rebuttal Period**
> > >
> > > Dear reviewer hiTa,
> > >
> > > We would like to thank you for spending your time carefully evaluating our submission, providing valuable feedback, and engaging in discussion with us.
> > >
> > > As the initial rebuttal period is expected to conclude soon, we hope our response addresses the concerns you raised.
> > >
> > > We look forward to hearing your feedback if you have any further requests for changes to the paper. We are happy to engage in further discussions during the following discussion period.
> > >
> > > We hope that you can reconsider your rating in light of our responses and changes to the paper.
> > >
> > > Best regards,
> > > Authors

---

### Official Review · Reviewer_4WVg · 2022-10-28

**Confidence:** 2
**Clarity, Quality, Novelty And Reproducibility:** The paper is well written. The propos…
**Correctness:** 3
**Technical Novelty And Significance:** 3
**Empirical Novelty And Significance:** 2
**Recommendation:** 6

**Strength And Weaknesses:**

Strength:
1. The authors propose to combine autoregression and deep learning, and train the network using variational inference.
2. The authors demonstrate that the proposed Rhino is structurally identifiable under similar assumptions to ANMs.
3. The paper is well written.



**Summary Of The Paper:**

In this paper, the authors investigate the problem of temporal causal discovery. Existing work relies on the absence of instantaneous effects with fixed noise distributions, whereas constraint-based methods require stronger faithfulness assumptions. To alleviate this problem, the authors propose Rhino, which combines vector auto-regression and deep learning, and is trained by variational inference. They further demonstrate that Rhino is structurally identifiable under similar assumptions to ANMs. Experiments are conducted on two discovery benchmarks.


**Summary Of The Review:**

The paper is well written. The proposed method is novel.

---

> ### Author Response · Authors · 2022-11-10
> **Author's response**
>
> We really appreciated your acknowledgement of our method. From the score in your review, it seems that our paper can still be improved. It would be great if you could give any specific points that you think can be improved or anything you have concerns with. That would be very helpful for us to improve the paper further.

---

### Official Review · Reviewer_w2Xd · 2022-10-28

**Confidence:** 3
**Clarity, Quality, Novelty And Reproducibility:** The work's quality needs some improve…
**Correctness:** 3
**Technical Novelty And Significance:** 3
**Empirical Novelty And Significance:** 3
**Recommendation:** 6

**Strength And Weaknesses:**

Strength
1. The problem causal discovery the paper is trying to tackle is quite important and interesting.
2. The method is supported by the structural identifiability theory.

Weakness
1. I do not understand why related works is not Section 2, why Section 5. This is quite confusing.
2. In the experiments section, I was wondering if we should only use AUC as the evaluation metrics since one major goal seems to be discover the causal relationships.

**Summary Of The Paper:**

The paper is trying to deal with non-linear relationships in time series data and proposes a structural equation model called Rhino, which combines vector auto-regression, deep learning as well as variational inference techniques.

**Summary Of The Review:**

Overall, I think the paper is of good quality, with some improvements required on the experiments and writing. But the overall the methodology and theory part is good.

---

> ### Author Response · Authors · 2022-11-10
> **Author's response**
>
> We really appreciate the feedback and your recognization of our contribution. Regarding your questions,
> ### Q1: Why related works are in Section 5
> Sections 2 and 5 serve two different purposes. Section 2 should not be treated as the literature review but as a prerequisite section for introducing the preliminary knowledge for building Rhino. Therefore we did not include many previous works. On the other hand, section 5 contains a summary of previous work without introducing the details. We have added their purposes in the revised paper.
>
> ### Q2: Additional metrics other than AUC for real-world experiments
> In the revised paper, we have added two additional metrics (Orientation F1 and SHD) for both DREAM3, and Netsim experiments in Appendix G. Both the newly added metrics agree with the trend of the original metric, AUROC, where our proposed method outperforms the baselines.
> The main reason we only reported AUROC for the two real-world experiments is for easy comparison. We directly cited many Granger causality baselines results from the previously published paper [1] for consistent comparisons. In addition, AUROC is also a commonly used metric for DREAM and Netsim datasets [2,3,4].
>
> ### References
> [1] Khanna, Saurabh, and Vincent YF Tan. "Economy statistical recurrent units for inferring nonlinear granger causality." arXiv preprint arXiv:1911.09879 (2019).
>
> [2] Tank, Alex, et al. "Neural granger causality." IEEE Transactions on Pattern Analysis and Machine Intelligence 44.8 (2021): 4267-4279.
>
> [3] Löwe, Sindy, et al. "Amortized causal discovery: Learning to infer causal graphs from time-series data." Conference on Causal Learning and Reasoning. PMLR, 2022.
>
> [4] Pamfil, Roxana, et al. "Dynotears: Structure learning from time-series data." International Conference on Artificial Intelligence and Statistics. PMLR, 2020.

---

> ### Author Response · Authors · 2022-11-17
> **End of Rebuttal Period**
>
> Dear reviewer w2Xd,
>
> We would like to thank you for spending your time carefully evaluating our submission and providing valuable feedback.
>
> As the initial rebuttal period is expected to conclude soon, we hope our response addresses the concerns you raised.
>
> We look forward to hearing your feedback if you have any further requests for changes to the paper. We would be happy to engage in further discussions during the following discussion period.
>
> We hope that you can reconsider your rating in light of our responses and changes to the paper.
>
> Best regards,
> Authors

---

### Author Response · Authors · 2022-11-10
**Summary of changes**

We really appreciate the valuable feedback from all the reviewers and the acknowledgement of our novelty and contributions. We have revised the paper according to the reviews. In the revised paper, we coloured the changes and used the labels "New", "Highlight" and "Revision" to indicate newly-added material, highlighting and revising the previous material, respectively. The question and reviewer labels are to indicate which specific feedback it targets. Here is the summary of the changes:
### Reviewer w2Xd:
- For Q1: Clarify the purpose of sections 2 and 5.
- For Q2: We reported additional metrics, orientation F1 and SHD, for both DREAM3 and Netsim datasets. Those metrics also agree with the AUROC.

### Reviewer hiTa:
- For Q1: We explicitly mention the proof technique adapted from the previous work.
- For Q2: We also clarify the definition of Rhino, which is the entire discovery framework, including the SEMs and variational inference algorithm.
- For Q3: We added a paragraph to mention that global optimum is required for Theorem 2 to hold.
- For Q5: We added the formal definition of "correctly specified model" in appendix B and add this reference to Theorem 2.

---

### Decision · Program_Chairs · 2023-01-20

**Decision:**

Accept: notable-top-25%

**Justification For Why Not Higher Score:**

Presentation and clarity of some of the results need improvement

**Justification For Why Not Lower Score:**

Interesting problem, solid method

**Metareview: Summary, Strengths And Weaknesses:**

The paper proposes a time-series structural equation model with non-linear lagged and instantaneous effects for a new structure identifiability result for temporal causal discovery. While existing work assumed independent noise, the new result here allows for history-dependent noise. The proposed structure identification method combines continuous optimization, variational inference, and deep neural networks. Empirical results on synthetic and real data show the applicability of the approach. The reviewers had concerns on the clarity and presentation on some of the results, most of which were addressed in the author response.

**Note From Pc:**

if the above contains the word "oral" or "spotlight" please see: "oral" presentation means -> notable-top-5% and "spotlight" means -> notable-top-25%. As stated in our emails, we are disassociating presentation type from AC recommendations